# ON THE BENEFITS OF MEMORY FOR MODELING TIME-DEPENDENT PDES

**Ricardo Buitrago Ruiz[1,2],**    **Tanya Marwah[1],**    **Albert Gu[1,2],**    **Andrej Risteski[1]**
[1]Carnegie Mellon University    [2]Cartesia AI

## ABSTRACT

Data-driven techniques have emerged as a promising alternative to traditional numerical methods for solving PDEs. For time-dependent PDEs, many approaches are Markovian—the evolution of the trained system only depends on the current state, and not the past states. In this work, we investigate the benefits of using memory for modeling time-dependent PDEs: that is, when past states are explicitly used to predict the future. Motivated by the Mori-Zwanzig theory of model reduction, we theoretically exhibit examples of simple (even linear) PDEs, in which a solution that uses memory is arbitrarily better than a Markovian solution. Additionally, we introduce Memory Neural Operator (MemNO), a neural operator architecture that combines recent state space models (specifically, S4) and Fourier Neural Operators (FNOs) to effectively model memory. We empirically demonstrate that when the PDEs are supplied in low resolution or contain observation noise at train and test time, MemNO significantly outperforms the baselines without memory—with up to $6\times$ reduction in test error. Furthermore, we show that this benefit is particularly pronounced when the PDE solutions have significant high-frequency Fourier modes (e.g., low-viscosity fluid dynamics) and we construct a challenging benchmark dataset consisting of such PDEs.

## 1 INTRODUCTION

Time-dependent partial differential equations (PDEs) are central to modeling various scientific and physical phenomena, necessitating the design of accurate and computationally efficient solvers. Recently, data-driven approaches based on neural networks (Li et al., 2024b; Lu et al., 2019) have emerged as an attractive alternative to classical numerical solvers, such as finite element and finite difference methods (LeVeque, 2007). Classical approaches are computationally expensive in high dimension and struggle with PDEs which are sensitive to initial conditions. Learned approaches can often negotiate these difficulties better, at least for the PDE family they are trained on.

One example of a data-driven approach is learning a *neural solution operator*, which for a time-dependent PDE learns to predict future states based on previous ones (Li et al., 2021; 2023a). Recent works (Tran et al., 2023; Lippe et al., 2023) suggest that optimal performance across various PDE families can be achieved by conditioning the models only on the immediate past state—i.e., treating the system as Markovian. In contrast, other works propose architectures that explicitly use memory of past states (Li et al., 2021; 2023a; Hao et al., 2024). However, none of these works elucidate whether and when modeling memory is helpful.

In this work, we demonstrate that when the solution of the PDE is only *partially observed* (e.g. observed at low resolution), explicitly modeling memory can be beneficial. Partial observability is natural in many practical settings. This could be due to limited resolution of the measurement devices collecting the data, inherent observational errors in the system, or prohibitive computational difficulty in generating high-quality synthetic data. This can lead to significant information loss, particularly in systems like turbulent flows (Pope, 2001) or shock formation in fluid dynamics (Christodoulou, 2007), where PDEs change abruptly in space and time. In such situations, classical results from dynamical systems (Mori-Zwanzig theory), suggest that the system becomes strongly non-Markovian.

More precisely, Mori-Zwanzig theory (Mori, 1965; Zwanzig, 1961; Ma et al., 2018) is an ansatz to understand the evolution of a subspace of a system (e.g., the span of the $k$ largest Fourier compo-

nents). Under certain conditions, this evolution can be divided into a *Markovian term* (the evolution of the chosen subspace under the PDE), a *memory term* (which is a weighted sum of the values of all previous iterates in the chosen subspace), and an *"unobservable" term*, which depends on the values of the initial conditions orthogonal to the selected subspace.

The main focus of this paper is studying *when* explicitly modeling this memory term is useful. We give an example of a very simple (in fact, linear) PDE where we show theoretically that the solution which takes into account the memory term can be arbitrarily better than the Markovian solution. We also provide a way to *operationalize* the Mori-Zwanzig formalism by introducing **Memory Neural Operator (MemNO)**, a neural operator architecture that combines a Markovian operator to model the spatial dynamics of the PDE (such as the Fourier Neural Operator (Li et al., 2021; Tran et al., 2023)), and a sequence model to maintain a compressed representation of the past states (such as the S4 state space model (Gu et al., 2022; 2023)). We show that MemNO outperforms its Markovian (memoryless) counterpart in PDEs observed on low resolution grids or with observation noise — achieving up to $6\times$ less test error. Our contributions are as follows:

- We identify a setting in which explicitly modeling memory is helpful: namely, when there is a combination of *lossy observations* of the solution of the PDE (e.g., due to limited resolution or observation noise) and significant contributions from *high-frequency Fourier modes* in the solution.

- Even in simple PDEs, we *theoretically* show the memory term can result in a solution that is (arbitrarily) closer to the correct solution, compared to the Markovian approximation —in particular when the operator describing the PDE "mixes" the observed and unobserved subspace.

- Across several families of one-dimensional and two-dimensional PDEs, we *empirically* demonstrate that when the input is supplied on a low-resolution grid, or contains observation noise, neural operators with memory outperform Markovian operators by a significant margin. More precisely, to *operationalize* memory, we introduce MemNO, a neural operator architecture combining Fourier Neural Operators (FNOs) and S4, which achieves the best performance across several Markovian and memory baselines.

- We observe that many current benchmarks for PDE solvers predominantly include PDEs in which there is little contribution from high-frequency Fourier modes. Consequently, we *construct more challenging datasets* where the solutions have significant high-frequency modes, which we believe will be of broader significance to the community beyond testing the effects of memory— especially given recent meta-studies suggesting many current PDE benchmarks are too easy (McGreivy & Hakim, 2024).

## 2  RELATED WORK

Data-driven neural solution operators (Chen & Chen, 1995; Bhattacharya et al., 2021; Lu et al., 2019; Kovachki et al., 2023) have emerged as the dominant approach for approximating PDEs, given their ability to model multiple families of PDEs at once, and their computational efficiency at inference time. Many architectures have been proposed to improve their performance across different families of PDEs: Li et al. (2021) introduced the Fourier Neural Operator (FNO), a resolution invariant architecture that uses a convolution-based integral kernel evaluated in the Fourier space; Tran et al. (2023) later introduced the Factorized FNO (FFNO) architecture, which builds upon and improves the FNO architecture by adding separable spectral layers and residual connections; Cao (2021) proposed a Transformer method with linear attention over the spatial sequence; other recent works have used U-Net-based architectures (Gupta & Brandstetter, 2023; Rahman et al., 2023).

Focusing on memory, Tran et al. (2023) performed ablations that suggest the Markov assumption is optimal and outperforms models that use the history of past timesteps as input. Lippe et al. (2023) performed a similar study for long rollouts of the PDE solution and concluded the optimal performance is indeed achieved under the Markovian assumption. We show that these findings can be replicated only when the resolution of the observation grid is high. On the other hand, we show that MemNO effectively models memory to achieve much superior performance than Markovian operators in low resolution, while not dropping performance in the high resolution case.

Our work is motivated by the Mori-Zwanzig formalism (Zwanzig, 1961; Mori, 1965) which shows that a partial observation of the current state of the system can be compensated using memory of

past states. Ma et al. (2018) draws parallels to the Mori-Zwanzig equations and LSTM (Hochreiter & Schmidhuber, 1997) to model the dynamics of the $k$ largest Fourier components of a single PDE. However, in our work, we study the benefits of memory in neural operators that learn the dynamics of an entire family of PDE. Furthermore, we show conditions under which not maintaining memory can result in arbitrarily large errors.

## 3 PRELIMINARIES

First, we introduce several definitions, as well as the Mori-Zwanzig formalism applied to our setting.

### 3.1 PARTIAL DIFFERENTIAL EQUATIONS (PDEs)

**Definition 1** (Space of square integrable functions). For integers $d$, $V$ and an open set $\Omega \subset \mathbb{R}^d$, we define $L^2\left(\Omega; \mathbb{R}^V\right)$ as the space of square integrable functions $u : \Omega \to \mathbb{R}^V$ such that $\|u\|_{L^2} \leq \infty$, where $\|u\|_{L^2} = \left(\int_\Omega \|u(x)\|_2^2 dx\right)^{\frac{1}{2}}$.

**Notation 1** (Restriction). Given a function $u : \Omega \to \mathbb{R}^V$ and a subset $A \subset \Omega$, we denote $u_{|A}$ as the restriction of $u$ to the domain $A$, i.e. $u_{|A} : A \to \mathbb{R}^V$.

The general form of the PDEs we consider in this paper will be the following:

**Definition 2** (Time-Dependent PDE). For an open set $\Omega \subset \mathbb{R}^d$ and an interval $[0, T] \subset \mathbb{R}$, a Time-Dependent PDE is the following expression:

$$\frac{\partial u}{\partial t}(t, x) = \mathcal{L}[u](t, x), \qquad \forall t \in [0, T], x \in \Omega, \tag{1}$$

$$u(0, x) = u_0(x), \qquad \forall x \in \Omega, \tag{2}$$

$$\mathcal{B}[u_{|\partial\Omega}](t) = 0, \qquad \forall t \in [0, T] \tag{3}$$

where $\mathcal{L} : L^2\left(\Omega; \mathbb{R}^V\right) \to L^2\left(\Omega; \mathbb{R}^V\right)$ is a differential operator in $x$ which is independent of time, $u_0(x) \in L^2\left(\Omega; \mathbb{R}^V\right)$ and $\mathcal{B}$ is an operator defined on the boundary of $\partial\Omega$, commonly referred to as the boundary condition.

In our theory and experiments, we will work with periodic boundary conditions (for a precise definition, see Definition 6). Finally, we will frequently talk about a grid of a given resolution:

**Definition 3** (Equispaced grid with resolution $f$). Let $\Omega = [0, L]^d$. An equispaced grid with resolution $f$ in $\Omega$ is the following set $\mathcal{S} \subset \mathbb{R}^d$:

$$\mathcal{S} = \left\{ \left(i_1 \frac{L}{f}, \cdots, i_k \frac{L}{f}\right) \middle| 0 \leq i_k \leq f - 1 \text{ for } 1 \leq k \leq d \right\}.$$

We will also denote by $|\mathcal{S}|$ the number of points in $\mathcal{S}$.

### 3.2 MORI-ZWANZIG FORMALISM

The Mori-Zwanzig formalism (Zwanzig, 2001) considers the setting in which an equation is known for a full system, but only a part of it is observed. It leverages the knowledge of past states of a system to compensate for the loss of information that arises from the partial observation of the current state. In our work, partial observation can refer to observing the solution at a discretized grid in space or only observing the Fourier modes up to a critical frequency. In the context of time-dependent PDEs, the Mori-Zwanzig principle is formalized as the *Nakajima–Zwanzig equation* (Nakajima, 1958).

We will give an overview of the Nakajima-Zwanzig equation and set up the notation for the rest of the paper. Assume we have a PDE as in Definition 2. Let $\mathcal{P} : L^2\left(\Omega; \mathbb{R}^V\right) \to L^2\left(\Omega; \mathbb{R}^V\right)$ be a linear projection operator. We define $\mathcal{Q} = I - \mathcal{P}$, where $I$ is the identity operator. In our setting, for the PDE solution at timestep $t$ $u_t \in L^2\left(\Omega; \mathbb{R}^V\right)$, $\mathcal{P}[u_t]$ is the part of the solution that we observe and $\mathcal{Q}[u_t]$ is the unobserved part. Thus, the initial information we receive for the system is $\mathcal{P}[u_0]$.

Applying $\mathcal{P}$ and $\mathcal{Q}$ to Equation 1 and using $u = \mathcal{P}[u] + \mathcal{Q}[u]$, we get:

$$\frac{\partial}{\partial t}\mathcal{P}[u](t,x) = \mathcal{P}\mathcal{L}[u](t,x) = \mathcal{P}\mathcal{L}\mathcal{P}[u](t,x) + \mathcal{P}\mathcal{L}\mathcal{Q}[u](t,x) \tag{4}$$

$$\frac{\partial}{\partial t}\mathcal{Q}[u](t,x) = \mathcal{Q}\mathcal{L}[u](t,x) = \mathcal{Q}\mathcal{L}\mathcal{P}[u](t,x) + \mathcal{Q}\mathcal{L}\mathcal{Q}[u](t,x) \tag{5}$$

Solving for 5 yields $\mathcal{Q}[u](t,x) = \int_0^t \exp\{\mathcal{Q}\mathcal{L}(t-s)\}\mathcal{Q}\mathcal{L}\mathcal{P}[u](s,x)ds + e^{\mathcal{Q}\mathcal{L}t}\mathcal{Q}[u_0](t,x)$.

Plugging into 4, we obtain a *Generalized Langevin Equation* (Mori, 1965) for $\mathcal{P}[u]$:

$$\frac{\partial}{\partial t}\mathcal{P}[u](t,x) = \mathcal{P}\mathcal{L}\mathcal{P}[u](t,x) + \mathcal{P}\mathcal{L}\int_0^t \exp\{\mathcal{Q}\mathcal{L}(t-s)\}\mathcal{Q}\mathcal{L}\mathcal{P}[u](s,x)ds + \mathcal{P}\mathcal{L}e^{\mathcal{Q}\mathcal{L}t}\mathcal{Q}[u_0](t,x) \tag{6}$$

We will refer to the first summand on the right hand side of Equation 6 as the **Markovian** term because it only depends on $\mathcal{P}[u](t,x)$, the second summand as the **memory** term because it depends on $\mathcal{P}[u](s,x)$ for $0 \leq s \leq t$, and the third summand as the **unobserved residual** as it depends on $\mathcal{Q}[u_0]$ which is never observed.

Since Equation 6 is exact, it is equivalent to solving the full system. The term that is typically most difficult to compute is the memory term, and many methods to approximate it have been proposed.

In the *physics literature*, some techniques include a perturbation expansion of the exponential $\exp\{\mathcal{Q}\mathcal{L}(t-s)\}$ (Breuer & Petruccione, 2002), or approximations using operators defined in $\mathcal{P}\left[L^2\left(\Omega; \mathbb{R}^V\right)\right]$ (Shi & Geva, 2003; Zhang et al., 2006; Montoya-Castillo & Reichman, 2016; Kelly et al., 2016). In the *classical numerical PDE solver* literature, the memory term has been approximated by leveraging the structure of the orthogonal dynamics of the $\mathcal{P}$ semi-group (Gouasmi et al., 2017), and the Mori-Zwanzig formalism has been applied to a variety of fluid dynamics PDEs (Parish & Duraisamy, 2017). In the *machine learning literature*, some works approximate the memory term with a neural network, which is then used as a part of a hybrid PDE solver (Ma et al., 2018; Beck et al., 2019; Pan & Duraisamy, 2018). Gupta & Lermusiaux (2021) approximated both the Markovian and memory term with a neural network, yet the method required deriving and coding adjoint equations to perform backpropagation. In this work, we explain *when* modeling memory is expected to be helpful, and introduce a neural operator that learns to model the temporal (e.g. memory) and spatial dynamics of a PDE directly from data.

## 4 THEORETICAL MOTIVATION FOR MEMORY: A SIMPLE EXAMPLE

In this section, we provide a simple, but natural example of a (linear) PDE, along with (in the nomenclature of Section 3.2) a natural projection operator given by a *Fourier truncation measurement operator*, such that the memory term in the generalized Langevin equation (GLE) can have an arbitrarily large impact on the quality of the calculated solution. We will work with periodic functions over $[0, 2\pi]$ which have a convenient basis:

**Definition 4** (Basis for $2\pi$-periodic functions). A function $f : \mathbb{R} \to \mathbb{R}$ is $2\pi$-periodic if $f(x+2\pi) = f(x)$. We can identify $2\pi$-periodic functions with functions over the torus $T := \{e^{i\theta} : \theta \in \mathbb{R}\} \subseteq \mathbb{C}$ by the map $\tilde{f}(e^{ix}) = f(x)$. Note that $\{e^{ixn}\}_{n \in \mathbb{Z}}$ is a basis for the set of $2\pi$-periodic functions.

We will define the following measurement operator:

**Definition 5** (Fourier truncation measurement). The operator $\mathcal{P}_k : L^2(T; \mathbb{R}) \to L^2(T; \mathbb{R})$ acts on $f \in L^2(T; \mathbb{R})$, $f(x) = \sum_{n=-\infty}^{\infty} a_n e^{inx}$ as $\mathcal{P}_k(f) = \sum_{n=-k}^{k} a_n e^{inx}$.

For notational convenience, we will also define the functions $\{\mathbf{e}_n\}_{n \in \mathbb{Z}}$, where $\mathbf{e}_n(x) := e^{-inx} + e^{inx}$. Now, we consider the following operator to define a linear time-dependent PDE:

**Proposition 1.** *Let* $\mathcal{L} : L^2(T; \mathbb{R}) \to L^2(T; \mathbb{R})$ *be defined as* $\mathcal{L}u(x) = -\Delta u(x) + B \cdot (e^{-ix} + e^{ix})u(x)$ *for* $B > 0$. *Then, we have:*

$$\forall 1 \leq n \in \mathbb{N}, \quad \mathcal{L}(\mathbf{e}_n) = n^2 \mathbf{e}_n + B(\mathbf{e}_{n-1} + \mathbf{e}_{n+1}) \quad \& \quad \mathcal{L}(\mathbf{e}_0) = 2B\mathbf{e}_1$$

The crucial property of this operator is that it acts by "mixing" the $n$-th Fourier basis with the $(n-1)$-th and $(n+1)$-th: thus information is propagated to both the higher and lower-order part of the spectrum. Given the above proposition, we can easily write down the evolution of a PDE with operator $\mathcal{L}$ in the basis $\{\mathbf{e}_n\}_{n\in\mathbb{Z}}$:

**Proposition 2.** *Let $\mathcal{L}$ be defined as in Proposition 1. Consider the PDE*

$$\frac{\partial}{\partial t}u(t,x) = \mathcal{L}u(t,x)$$

$$u(0,x) = \sum_{n\in\mathbb{N}_0} a_n(0)\mathbf{e}_n$$

*Let $u(t,x) = \sum_{n\in\mathbb{N}_0} a_n^{(t)}\mathbf{e}_n$. Then, the coefficients $a_n^{(t)}$ satisfy:*

$$\forall 1 \le n \in \mathbb{N}, \ \frac{\partial}{\partial t}a_n^{(t)} = n^2 a_n^{(t)} + B\left(a_{n-1}^{(t)} + a_{n+1}^{(t)}\right) \tag{7}$$

$$\frac{\partial}{\partial t}a_0^{(t)} = 2Ba_1^{(t)} \tag{8}$$

With this setup in mind, we will show that as $B$ grows, the memory term in Equation 6 can have an arbitrarily large effect on the calculated solution:

**Theorem 1** (Effect of memory). *Consider the operator $\mathcal{L}$ defined in Proposition 1, the Fourier truncation operator $\mathcal{P}_1$, and let $\mathcal{Q} = I - \mathcal{P}_1$. Let $u(0,x)$ have the form in Proposition 2 for $B > 0$ sufficiently large, and let $a_n^{(0)} > 0, \forall n > 0$. Consider the memoryless and memory-augmented PDEs:*

$$\frac{\partial u_1}{\partial t} = \mathcal{P}_1 \mathcal{L}u_1 \tag{9}$$

$$\frac{\partial u_2}{\partial t} = \mathcal{P}_1 \mathcal{L}u_2 + \mathcal{P}_1 \mathcal{L}\int_0^t \exp\{\mathcal{Q}\mathcal{L}(t-s)\}\mathcal{Q}\mathcal{L}u_2(s)ds \tag{10}$$

*with $u_1(0,x) = u_2(0,x) = \mathcal{P}_1 u(0,x)$. Then, $u_1$ and $u_2$ satisfy:*

$$\forall t > 0, \|u_1(t) - u_2(t)\|_{L_2} \gtrsim Bt\|u_1(t)\|_{L_2} \tag{11}$$

$$\forall t > 0, \|u_1(t) - u_2(t)\|_{L_2} \gtrsim Bt\exp\left(\sqrt{2}Bt\right) \tag{12}$$

**Remark 1.** *Note that the two conclusions of the theorem mean that both the absolute difference, and the relative difference between the PDE including the memory term Equation 10 and not including the memory term Equation 9 can be arbitrarily large as $B, t \to \infty$.*

**Remark 2.** *The choice of $\mathcal{L}$ is made for ease of calculation of the Markovian and memory term. Conceptually, we expect the solution to Equation 10 will differ a lot from the solution to Equation 9 if the action of the operator $\mathcal{L}$ tends to "mix" components in the span of $\mathcal{P}$ and the span of $\mathcal{Q}$.*

**Remark 3.** *If we solve the equation $\frac{\partial}{\partial t}u(t,x) = \mathcal{L}u(t,x)$ exactly, we can calculate that $\|u(t)\|_{L_2}$ will be on the order of $\exp(2Bt)$. This can be seen by writing the evolution of the coefficients of $u(t)$ in the basis $\{\mathbf{e}_n\}$, which looks like: $\frac{\partial}{\partial t}\begin{pmatrix} a_0 \\ a_1 \\ \dots \end{pmatrix} = \mathcal{O}\begin{pmatrix} a_0 \\ a_1 \\ \dots \end{pmatrix}$ where $\mathcal{O}$ is roughly a tridiagonal*

*Toeplitz operator $\mathcal{O} = \begin{pmatrix} \vdots & \vdots & \vdots & \vdots & \\ \dots & B & n^2 & B & 0 & \dots \\ \dots & 0 & B & (n+1)^2 & B & \dots \\ \vdots & \vdots & \vdots & \vdots & \end{pmatrix}$. The largest eigenvalue of this oper-*

*ator can be shown to be on the order of at least $2B$ (Equation 4 in Noschese et al. (2013)). The Markovian term results in a solution of order $\exp(\sqrt{2}Bt)$ ( Equation 19 and Equation 20), which is multiplicatively smaller by a factor of $\exp((2-\sqrt{2})Bt)$. The result in this Theorem shows that the memory-based PDE Equation 10 results in a multiplicative "first order" correction which can be seen by Taylor expanding $\exp(\sqrt{2}Bt) \approx 1 + \sqrt{2}Bt + \frac{1}{2}(\sqrt{2}B)^2t^2 + \dots$.*

## 5 EXPERIMENTAL SETUP

### 5.1 DATASET GENERATION

**PDEs with high-frequency Fourier modes:** From the expression for the memory term in Equation 6 and the presence of high-frequency terms in the solution of the PDE of Theorem 1, we should intuitively expect that memory will be most useful when the PDE solutions contain significant contributions from high-frequency Fourier modes[1]. Nevertheless, current benchmarks like PDEBench (Takamoto et al., 2023) rarely contain PDEs whose solutions have substantial high-frequency components, as we quantitatively show in Appendix D. A solution which predominantly contains low-frequency Fourier modes can be accurately approximated by its Fourier truncation (Definition 5), so it can be represented by a finite-dimensional space, which implies that the unobserved part of the solution ($\mathcal{Q}[u]$ in the notation of Section 3.2) should be small.

Therefore, we construct a new benchmark dataset which is specifically designed to contain PDEs in which the high-frequency Fourier modes have substantial contribution. Specifically, we generate a benchmark from solutions to the Kuramoto-Sivashinsky equation with low viscosity (Section 6.1). In the case of Navier-Stokes (Section 6.2) and Burgers' equation (Section C), we directly take datasets from previous works. Details on data generation procedure are provided in Appendix E.

**Datasets with different resolutions:** To construct our datasets, we first take discretized trajectories of a PDE on a *high resolution* discretized spatial grid $\mathcal{S}^{HR} \subset \mathbb{R}^d$, i.e. $u(t) \in \mathbb{R}^{|\mathcal{S}^{HR}|}$. We then produce datasets that consist of *lower resolution* versions of the above trajectories, i.e. on a grid $\mathcal{S}^{LR}$ of lower resolution $f$, and show the performance of models that were trained and tested at such resolution. For 1-dimensional datasets, the discretized trajectory on $\mathcal{S}^{LR}$ is obtained by cubic interpolation of the trajectory in the highest resolution grid. In 2D, the discretized trajectory is obtained by downsampling.

### 5.2 TRAINING AND EVALUATION PROCEDURE

**Task:** Let $u \in \mathcal{C}\left([0, T]; L^2\left(\Omega; \mathbb{R}^V\right)\right)$ be the solution of the PDE given by Definition 2. Let $\mathcal{S}$ be an equispaced grid in $\Omega$ with resolution $f$, and let $\mathcal{T}$ be another equispaced grid in $[0, T]$ with $N_t + 1$ points. Given $u_0(x)_{|\mathcal{S}}$, our goal is to predict $u(t, x)_{|\mathcal{S}}$ for $t \in \mathcal{T}$ using a neural operator.

**Training objective:** As is standard, we proceed by empirical risk minimization on a dataset of trajectories. More specifically, given a loss function $\ell : \left(\mathbb{R}^{|\mathcal{S}|}, \mathbb{R}^{|\mathcal{S}|}\right) \to \mathbb{R}$, a dataset of training trajectories $\left(u(t, x)^{(i)}\right)_{i=0}^N$, and parametrized maps $\mathcal{G}_t^\Theta : \mathbb{R}^{|\mathcal{S}|} \to \mathbb{R}^{|\mathcal{S}|}$ for $t \in \mathcal{T}$, we optimize:

$$\Theta^* = \text{argmin}_\Theta \frac{1}{N} \sum_{i=0}^{N-1} \frac{1}{N_t} \sum_{t=1}^{N_t} \ell\left(u(t, x)^{(i)}_{|\mathcal{S}}, \mathcal{G}_t^\Theta\left[u_0^{(i)}(x)_{|\mathcal{S}}\right]\right)$$

**Training and evaluation metric:** Our training loss and evaluation metric is *normalized Root Mean Squared Error (nRMSE)*:

$$\text{nRMSE}\left(u(t, x)_{|\mathcal{S}}, \hat{u}(t)\right) := \frac{\|u(t, x)_{|\mathcal{S}} - \hat{u}(t)\|_2}{\|u(t, x)_{|\mathcal{S}}\|_2},$$

where $\| \cdot \|_2$ is the Euclidean norm in $\mathbb{R}^{|\mathcal{S}|}$.

Further details on training hyperparameters are given in Appendix F.

### 5.3 ARCHITECTURE FRAMEWORK: MEMORY NEURAL OPERATOR

In this section we describe Memory Neural Operator (MemNO), a deep learning framework to incorporate memory into neural operators. A diagram is provided in Figure 9 and pseudocode in Figure 10.

---

[1]Note, this is meant to be an intuitive rule-of-thumb rather than a formal statement. In general, the "observation" operator and the PDE will interact in complicated ways, but the combination of low-resolution grids and examining high-frequency components in the Fourier basis seems to be very predictive in practice.

Let $\text{NO}_t^\Theta$ be a neural operator with $L$ layers, and denote $\text{NO}_t^\Theta[u_0]$ the prediction of the solution of the PDE at time $t$. We will assume that this Neural Operator follows the Markovian assumption, i.e. we can write:

$$\text{NO}_{t_{i+1}}^\Theta[u_0] = r_{\text{out}} \circ \ell_L \circ \ell_{L-1} \circ ... \circ \ell_0 \circ r_{\text{in}}[\text{NO}_{t_i}^\Theta[u_0]], \tag{13}$$

where $r_{\text{in}} : \mathbb{R}^{|\mathcal{S}|} \to \mathbb{R}^{|\mathcal{S}| \times h_0}$ is the encoder and $r_{\text{out}} : \mathbb{R}^{|\mathcal{S}| \times h_{L+1}} \to \mathbb{R}^{|\mathcal{S}|}$ is the decoder; $\ell_j : \mathbb{R}^{|\mathcal{S}| \times h_j} \to \mathbb{R}^{|\mathcal{S}| \times h_{j+1}}$ are parametrized layers; and $h_j$ is the dimension of the $j$-th hidden layer. Essentially, the solution for each new timestep is obtained by applying the *same* $L$ layers to the immediately previous predicted timestep.

Our goal is to define a network $\mathcal{G}_t^\Theta$ that builds upon $\text{NO}_t^\Theta$ and can incorporate memory. For this, we take inspiration from the Mori-Zwanzig formalism summarized in Section 3.2. Comparing Equation 13 with Equation 6, we identify $\ell_L \circ \ell_{L-1} \circ ... \circ \ell_0$ with the Markov term which models the spatial dynamics. To introduce the memory term, we interleave an additional residual sequential layer $\mathcal{M}$ that acts on hidden representations of the solution at previous timesteps. Concretely, the MemNO architecture can be written as:

$$\mathcal{G}_{t_{i+1}}^\Theta[u_0] = r_{\text{out}} \circ \ell_L \circ ... \circ \ell_{k+1} \circ \mathcal{M} \circ \ell_k \circ ... \circ \ell_0 \circ r_{\text{in}}\left[\mathcal{G}_{t_i}^\Theta[u_0], \mathcal{G}_{t_{i-1}}^\Theta[u_0], ..., u_0\right],$$

where $-1 \le k \le L$ is a chosen hyperparameter.[2] For notation, we will refer to $v^{(j)}(t') \in \mathbb{R}^{|S| \times h_j}$ as the hidden representation at the $j$-th layer for a timestep $t' \le t_i$, and $v^{(j)}(t', x) \in \mathbb{R}^{h_j}$ as the value of such hidden representation at a spatial point $x \in \mathcal{S}$. Then, the spatial $\ell_j$ layers are understood to be applied timestep-wise, i.e. $\ell_j\left[v^{(j)}(t_i), ..., v^{(j)}(t_0)\right] \coloneqq \left[\ell_j[v^{(j)}(t_i)], ..., \ell_j[v^{(j)}(t_0)]\right]$, and analogously for $r_{\text{in}}$ and $r_{\text{out}}$. Thus, the $\ell_j$ layers still follow the Markovian assumption. The memory is introduced through $\mathcal{M}$, which is a sequential model that uses the history of the previous timesteps to predict the next one. For computational efficiency, we consider a sequential model $\mathcal{M} : \mathbb{R}^{i \times h_k} \longrightarrow \mathbb{R}^{h_k}$ that is applied to each element of the spatial dimension $|\mathcal{S}|$ independently, i.e. for each $x \in \mathcal{S}$, $\left(\mathcal{M}[v^{(k)}(t_i), ..., v^{(k)}(t_0)]\right)(x) \coloneqq \mathcal{M}[v^{(k)}(t_i, x), ..., v^{(k)}(t_0, x)]$.[3]

Note that our MemNO framework can be combined with *any* existing neural operator layer $\ell$, and with any (causal) sequential model $\mathcal{M}$. Thus it provides a modular architecture design framework which we hope can serve as a useful tool for practitioners.

## 5.4 Instantiating the Memory Neural Operator framework: S4FFNO

For our experiments, we introduce S4 Factorized Fourier Neural Operator (S4FFNO), which instantiates the MemNO framework by combining the Factorized Fourier Neural Operator (FFNO) (Tran et al., 2023) as the Markovian neural operator and S4 (Gu et al., 2022) as the sequential layer. We choose S4 models over recurrent architectures like LSTM (Hochreiter & Schmidhuber, 1997) due to superior performance in modeling long range dependencies (Gu et al., 2022; Tay et al., 2020), ease of training, and favorable memory and computational scaling with both state dimension and sequence length. An ablation comparing S4 to LSTM and Transformers is provided in Appendix G.1.

## 6 Memory helps in low-resolution and input noise: a case study

In this section we present a case study for several PDEs of practical interest, showing that neural operators with memory confer accuracy benefits when the data is supplied in low resolution or with observation noise. We will use four Markovian baselines: **Factformer (1D)** (Li et al., 2023b), The Galerkin Transformer **(GKT)** (Cao, 2021), the U-Net Neural Operator **(U-Net)** (Gupta & Brandstetter, 2023), and the Factorized Fourier Neural Operator **(FFNO)** (Tran et al., 2023). For a memory-augmented baseline, we consider the Multi Input Factorized Fourier Neural Operator **(Multi input FFNO)**, which takes as input the last 4 timesteps of the solution of the PDE to predict the next one, as proposed in the original FNO paper (Li et al., 2021), yet using the architectural design of FFNO. The architectural details for all the models are elaborated upon in Appendix B.

---

[2]$k = L$ refers to inserting $M$ after all the $S$ layers, and $k = -1$ refers to inserting $M$ as the first layer. In Appendix G.2, we show our experiments are not very sensitive to the choice of $k$.

[3]We present an analysis on some architecture modifications that model the local spatial structure more explicitly in Appendix I.

| Architecture | Uses memory | Resolution | nRMSE ↓ | | | |
| --- | --- | --- | --- | --- | --- | --- |
| | | | KS | | | Burgers' |
| | | | $\nu = 0.075$ | $\nu = 0.1$ | $\nu = 0.125$ | $\nu = 0.001$ |
| Factformer (1D) | ✗ | | 0.436 | 0.391 | 0.149 | 0.190 |
| GKT | ✗ | | 0.588 | 0.601 | 0.314 | 0.356 |
| U-Net | ✗ | 32 | 0.542 | 0.511 | 0.249 | 0.188 |
| FFNO | ✗ | | 0.500 | 0.446 | 0.187 | 0.207 |
| Multi Input FFNO | ✓ | | 0.364 | 0.308 | 0.092 | 0.099 |
| S4FFNO (Ours) | ✓ | | **0.139** | **0.108** | **0.031** | **0.053** |
| Factformer (1D) | ✗ | | 0.195 | 0.086 | 0.022 | 0.162 |
| GKT | ✗ | | 0.401 | 0.120 | 0.016 | 0.349 |
| U-Net | ✗ | 64 | 0.147 | 0.062 | 0.022 | 0.171 |
| FFNO | ✗ | | 0.107 | 0.033 | **0.004** | 0.146 |
| Multi Input FFNO | ✓ | | 0.108 | 0.046 | 0.005 | 0.054 |
| S4FFNO (Ours) | ✓ | | **0.036** | **0.011** | **0.004** | **0.037** |
| Factformer (1D) | ✗ | | 0.058 | 0.030 | 0.017 | 0.117 |
| GKT | ✗ | | 0.028 | 0.013 | 0.007 | 0.307 |
| U-Net | ✗ | 128 | 0.033 | 0.027 | 0.014 | 0.112 |
| FFNO | ✗ | | **0.006** | **0.004** | **0.002** | 0.099 |
| Multi Input FFNO | ✓ | | 0.057 | 0.052 | 0.023 | **0.028** |
| S4FFNO (Ours) | ✓ | | 0.008 | 0.005 | 0.003 | 0.030 |

Table 1: nRMSE values at different resolutions for Burgers' and KS with different viscosities. S4FFNO achieves up to 6x less error than its memoryless counterpart (FFNO) in KS at resolution 32. The final time of KS is 2.5 seconds and it contains 25 timesteps. The final times of Burgers' is 1.4 seconds and it contains 20 timesteps. For the prediction at time $t$, S4FFNO has access to the (compressed) memory of all previous timesteps, whereas Multi Input FFNO takes as input the previous four timesteps. More details on training are given in Appendix F, and on the Burgers' experiment in Appendix C.

## 6.1 KURAMOTO–SIVASHINSKY EQUATION (1D): STUDY IN LOW-RESOLUTION

The Kuramoto-Sivashinsky equation (KS) is a nonlinear PDE that is used as a modeling tool in fluid dynamics, chemical reaction dynamics, and ion interactions. Due to its chaotic behavior it can model instabilities in various physical systems. For viscosity $\nu \in \mathbb{R}_+$, it is written as $u_t + uu_x + u_{xx} + \nu u_{xxxx} = 0$. We generated datasets for KS at different viscosities and resolutions, and show the results in Table 1. At resolutions 32 and 64, the memory models (S4FFNO and Multi Input FFNO) outperform the Markovian baselines. In particular, S4FFNO can achieve up to $6\times$ less error than its Markovian counterpart (FFNO) and additionally $3\times$ less error than Multi Input FFNO. Furthermore, in Appendix H.2 we show that S4FFNO still achieves 3x less error than FFNO models with 7x more parameters.

At resolution 128, FFNO has similar performance compared to S4FFNO, and it outperforms Multi Input FFNO. This is in agreement with other works which propose following the Markovian assumption in neural operators (Tran et al., 2023; Lippe et al., 2023), where it is argued that incorporating previous timesteps as input is not necessary and can lead to difficulties in learning, as it seems to happen with Multi Input FFNO. By contrast, S4FFNO effectively models memory when it is useful (resolutions 32 and 64) without compromising performance at higher resolutions.

In Figure 1 we show the performance of all models across a continuous range of resolutions. It can be seen that there is a "cutoff" resolution at which memory models (i.e. S4FFNO) start outperforming Markovian ones (i.e. FFNO) by a large margin. Very importantly, this cutoff resolution depends on the viscosity, being around 76 when $\nu = 0.075$, 68 when $\nu = 0.1$, and 52 when $\nu = 0.125$. In the KS equation, a lower viscosity leads to the appearance of higher frequencies in the Fourier spectrum (see second row of Figure 1), which are not well captured at low resolutions. Thus, we identify the *resolution relative to the Fourier frequency spectrum of the solution* as a key factor for the improved performance of MemNO over memoryless neural operators. We note that even if the initial condition does not contain high frequencies, in the KS equation high frequencies will appear as the system evolves. We provide a similar study on 1D Burgers equation in Appendix C.

Lastly, there are several architecture choices to model memory that improve performance in low resolution. In particular, in Appendix G.1 we show that using LSTM instead of S4 also brings similar

performance improvements. Likewise, in Appendix G.3 we show that using S4 as the memory model with U-Net as the Markovian neural operator also outperforms the purely Markovian U-Net.

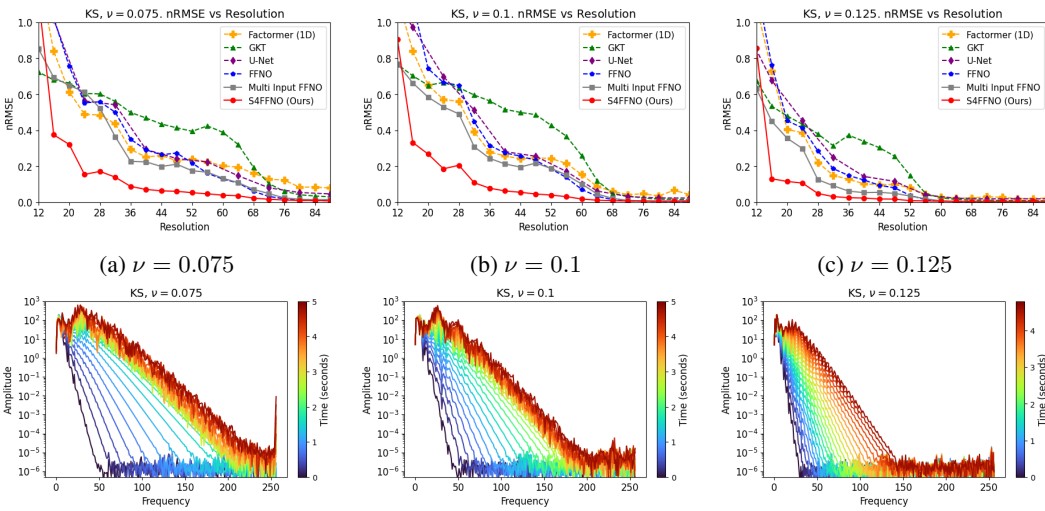

Figure 1: (First row) nRMSE for several models in the KS dataset at different resolutions, where each column is a different viscosity. The final time is $T = 2.5s$ and there are $N_t = 25$ timesteps. (Second row) A visualization of the whole frequency spectrum at each of the 25 timesteps for a single trajectory in the dataset. The spectrum is obtained with the ground truth solution at resolution 512.

## 6.2 NAVIER-STOKES EQUATION (2D): STUDY IN OBSERVATION NOISE

The Navier-Stokes equation describes the motion of a viscous fluid. Like in Li et al. (2021), we consider the incompressible form in the 2D unit torus, which is given by:

$$\frac{\partial w(x,t)}{\partial t} + u(x,t) \cdot \nabla w(x,t) = \nu \Delta w(x,t) + f(x), \qquad x \in (0,1)^2, t \in (0,T]$$
$$\nabla \cdot u(x,t) = 0, \qquad x \in (0,1)^2, t \in [0,T]$$
$$w(x,0) = w_0(x), \qquad x \in (0,1)^2$$

Where $w = \nabla \times u$ is the vorticity, $w_0 \in L^2((0,1)^2; \mathbb{R})$ is the initial vorticity, $\nu \in \mathbb{R}_+$ is the viscosity coefficient, and $f \in L^2((0,1)^2; \mathbb{R})$ is the forcing function. In general, the lower the viscosity, the more rapid the changes in the solution and the harder it is to solve it numerically or with a neural operator. We investigate the effect of memory when adding i.i.d. Gaussian noise to the inputs of our neural networks. The noise is sampled i.i.d. from a Gaussian distribution $\mathcal{N}(0, \sigma)$, and then added to training and test inputs. During training, for each trajectory a different noise (with the same $\sigma$) is sampled at each iteration of the optimization algorithm. The targets in training and testing represent our ground truth, and do not contain added noise. In Figure 2a, we show the results for $\nu = 10^{-3}$ when adding noise levels from $\sigma = 0.0$ (no noise) to $\sigma = 2.048$. S4FFNO-2D outperforms FFNO-2D across most noise levels, and the difference between the two is especially significant for noise levels beyond $0.128$, where FFNO-2D is around $50\%$ higher than S4FFNO-2D (note the logarithmic scale). For this viscosity, adding small levels of noise actually helps training, which was also observed in other settings in Tran et al. (2023). Figure 2b shows the same experiment performed with $\nu = 10^{-5}$. Again, S4FFNO-2D outperforms FFNO-2D across most noise levels. FFNO-2D losses are similarly around $50\%$ higher for noise levels above $0.032$. In this viscosity, adding these levels of noise does not help performance.

## 6.3 RELATIONSHIP WITH FRACTION OF UNOBSERVED MODES

In this section, we provide a simple experiment to quantify the effect of the fraction of unobserved modes on the performance of memory based models. Precisely, suppose $u \in L^2(\Omega; \mathbb{R}^V)$ is the

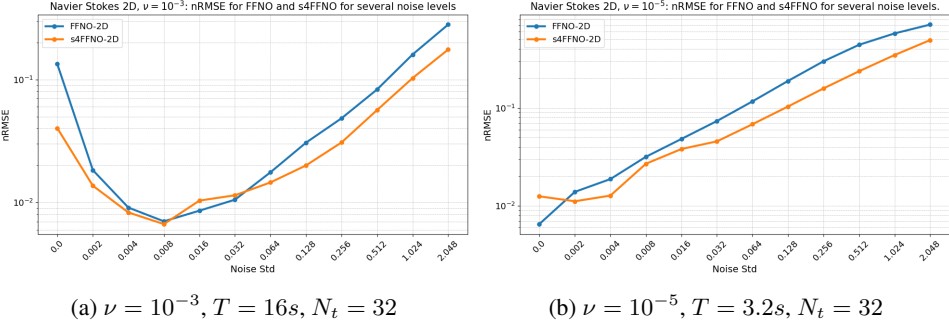

(a) $\nu = 10^{-3}$, $T = 16s$, $N_t = 32$      (b) $\nu = 10^{-5}$, $T = 3.2s$, $N_t = 32$

Figure 2: nRMSE of FFNO-2D and S4FFNO-2D trained on Navier-Stokes 2D with different noise standard deviations $\sigma$ added to training and test inputs. Two configurations of viscosity $\nu$ and final time $T$ are shown.

solution of a 1-dimensional PDE at a certain timestep, and $a_n$ for $n \in \mathbb{Z}$ is its Fourier Transform. If we observe it at a resolution f, we can only estimate its top $\lfloor \frac{f}{2} \rfloor$ modes[4]. Thus, we define $\omega_f$ as the ratio of unobserved modes at resolution $f$:

$$\omega_f := \frac{\sum_{|n| > \lfloor \frac{f}{2} \rfloor} |a_n|^2}{\sum_{n \in \mathbb{Z}} |a_n|^2} \tag{14}$$

$\omega_f$ is an approximate indicator of the amount of information that is lost when the solution of the PDE is observed at resolution $f$. In practice, $\omega_f$ can be computed by approximating the $a_n$ with the discrete Fourier modes of the solution in the highest resolution available. We show that there is a positive correlation between $\omega_f$ and the difference in nRMSE between FFNO and S4FFNO for the KS experiment in Figure 3, and also the for Burgers' experiments of Appendix C in Figure 5. This demonstrates the benefits of memory as a way to compensate for missing information in the observations.

## 7 CONCLUSION AND FUTURE WORK

We study the benefits of maintaining memory while modeling time dependent PDE systems. When we only observe part of the initial conditions (for example, PDEs observed on low-resolution or with input noise), the system is no longer Markovian, and the dynamics depend on a *memory term*. Taking inspiration from the Mori-Zwanzig formalism, we introduce MemNO, an architecture that combines Fourier Neural Operators (FNO) to model the spatial dynamics of the PDE, and the S4 sequence model to incorporate memory of past states. Through our experiments on different 1D and 2D PDEs, we show that the MemNO architecture outperforms the memoryless baselines, particularly when the solution to the PDE has large components on high-frequency Fourier modes.

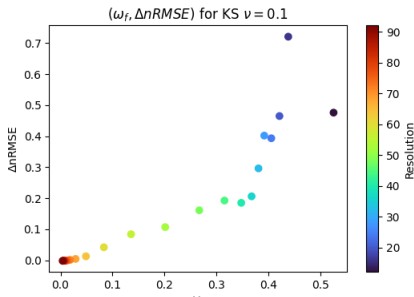

Figure 3: Values of $\omega_f$ and the difference in nRMSE between FFNO and S4FFNO for different resolutions in the KS experiment of Section 6.1 with $\nu = 0.1$. $\omega_f$ is averaged across all trajectories in the dataset and across all timesteps.

We present several avenues for future work. First, our experiments on observation noise are limited to the setting where the input noise is i.i.d. Further, extending the experiments and observing the effects of memory in more real-world settings (for example, with non-i.i.d. noise or in the presence of aliasing) seems fertile ground for future work, and also necessary to ensure that the application of this method does not have unintended negative consequences when broadly applied in society. Lastly, while we primarily compare between Markovian and memory architectures, a study on the trade-offs between different memory architectures such as S4FFNO, LSTM-FFNO, S4U-Net and Multi Input FFNO is an interesting direction for future work.

---

[4]This is a consequence of the Nyquist–Shannon sampling theorem.

ACKNOWLEDGEMENTS

RBR is supported by the "la Caixa" Foundation (ID 100010434). The fellowship code is LCF/BQ/EU22/11930090. TM is supported in part by CMU Software Engineering Institute via Department of Defense under contract FA8702-15-D-0002. AR is supported in part by NSF awards IIS-2211907, CCF-2238523, and Amazon Research.

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

# A  ADDITIONAL RELATED WORK

**Neural Operators**. The Fourier Neural Operator (FNO) is a neural operator that performs a transformation in the frequency space of the input (Li et al., 2021). Other models have proposed different inductive biases for neural operators, including physics based losses and constraints (Li et al., 2024b), using Deep Equilibrium Model (DEQ) (Bai et al., 2019) to design specialized architectures for steady-state (time-independent) PDEs (Marwah et al., 2023), and using local message passing Graph Neural Networks (GNNs) (Gilmer et al., 2017; Kipf & Welling, 2016) encoders to model irregular geometries (Li et al., 2020; 2024a). Other methodologies to solve PDEs include methods like (Gupta & Brandstetter, 2023; Rahman et al., 2023) that use the U-Net (Ronneberger et al., 2015) architectures and works like Cao (2021); Hao et al. (2023) that introduce different Transformer-based (Vaswani et al., 2017) neural solution operators for modeling both time-dependent and time-independent PDEs. While most of these methodologies are designed for time-dependent PDEs, there is no clear consensus of how to use the past states to predict future states, and most of these methods predict the PDE states over time in an autoregressive way by conditioning the model on varying lengths of the past states (Li et al., 2021; Tran et al., 2023; Hao et al., 2023).

**Foundation models**. There have been community efforts towards creating large-scale foundational models for modeling diverse PDE families (McCabe et al., 2023; Hao et al., 2024; Shen et al., 2024), and weather prediction (Pathak et al., 2022; Lam et al., 2022).

# B  NETWORK ARCHITECTURES

For all our models, we use a simple spatial positional encoding $E$. In 1-D, if the grid has $f$ equispaced points in $[0, L]$, then $E \in \mathbb{R}^f$ and the positional encoding is defined as $E_i = \frac{i}{L}$ for $0 \le i \le f - 1$. In 2-D, if we have $f \times f$ points in a 2-D equispaced grid in $[0, L_x] \times [0, L_y]$, the positional encoding is defined as $E_{ij} = (\frac{i}{L_x}, \frac{j}{L_y})$. The input lifting operator (i.e. encoder) $r_{\text{in}}$ is a linear layer that maps a concatenation of input and grid to the hidden dimension $\mathbb{R}^2 \to \mathbb{R}^h$, which is applied to each element of the spatial dimension independently. It is shared across all model architectures. Likewise, for the decoder $\mathcal{R}_{\text{out}}$, we use another linear layer $\mathbb{R}^h \to \mathbb{R}$.

**Factorized Fourier Neural Operator (FFNO)** (Tran et al., 2023): This model is a refinement over the original Fourier Neural Operator (Li et al., 2021). Given a hidden dimension $h$ and a spatial grid $\mathcal{S}$, its layers $\ell : \mathbb{R}^{|\mathcal{S}| \times h} \to \mathbb{R}^{|\mathcal{S}| \times h}$ are defined as:

$$\ell(v) := v + \text{Linear}_{h,h'} \circ \sigma \circ \text{Linear}_{h',h} \circ \mathcal{K}[v] \tag{15}$$

where $\sigma$ is the GeLU activation function (Hendrycks & Gimpel, 2016) and $h'$ is an expanded hidden dimension. $\mathcal{K}$ is a kernel integral operator that performs a linear transformation in the frequency space. Denoting by $\text{FFT}_\alpha$, $\text{IFFT}_\alpha$ the Discrete Fast Fourier Transform and the Discrete Inverse Fast Fourier Transform along dimension $\alpha$ (Cooley et al., 1969) respectively, the operator can be written as:

$$\mathcal{K}[v] := \sum_{\alpha \in \{1, \dots, d\}} \text{IFFT}[R_\alpha \cdot \text{FFT}_\alpha[v]] \tag{16}$$

for learnable matrices of weights $R_\alpha \in \mathbb{C}^{h^2 \times k_{\max}}$. $k_{\max}$ is the maximum number of Fourier modes which are used in $\mathcal{K}$. We use all Fourier modes by setting $k_{\max} = \lfloor \frac{f}{2} \rfloor$.

In our experiments, The FFNO model consists of 4 FFNO layers. For experiments in 1D, the hidden dimensions are all 128 ($h_j = 128$ for $j = 0, 1, 2, 3$) and the expanded hidden dimension of FFNO's MLP $h'$ is $4 \cdot 128$. For experiments in 2D, the hidden dimensions are all 64 and the expanded hidden dimension is $4 \cdot 64$.

**S4 - Factorized Fourier Neural Operator (S4FFNO)**: This model uses our MemNO framework. To isolate the effect of memory, all layers except the memory layer are the same as FFNO. For the memory layer, we choose an S4 layer (Gu et al., 2022) with a state dimension of 64 and a diagonal S4 (S4D) kernel.[5]

---

[5]The S4 repository has two available kernels, the diagonal S4 (S4D) and the Normal Plus Low Rank S4 (S4NPLR). In our experiments, we didn't find a significant difference between the two, and chose S4D for simplicity.

**Multi Input Factorized Fourier Neural Operator (Multi Input FFNO)**: This architecture uses the solution at the last $K = 4$ timesteps as input to predict the next timestep, as originally proposed by Li et al. (2021). Thus, this model uses the (uncompressed) memory of the four previous timesteps and it is not Markovian. We choose $K = 4$ because, in practice, the number of previous timesteps to which we have access is limited, if any. We also believe that Multi Input FFNO is advantaged by having access to four ground truth observations, whereas the rest of the models only have access to one. Thus, we consider $K = 4$ to be a reasonable choice when considering practical applicability and fairness in comparisons. On the implementation side, the only difference with the FFNO architecture resides in the input lifting operator $\mathcal{R}_{in}$, which takes a concatenation of $u_{t_{i-3}}, u_{t_{i-2}}, u_{t_{i-1}}, u_{t_i}$ as input to predict $u_{t_{i+1}}$. In all our experiments, we choose the fourth timestep of the solution of the PDEs as initial condition for the rest of the models, whereas Multi Input FFNO is given access to the first, second, third, and fourth timesteps for its first prediction. The number of layers and hidden dimensions are the same as FFNO.

**Factformer 1D (Li et al., 2023b)**: This models uses four linear attention layers over the spatial sequence length and an MLP as output projection. We set the hidden dimension to 64, and each attention layer has 4 heads with a hidden dimension of 128, thus expanding the dimension from 64 to 512. The implementation is taken from `https://github.com/BaratiLab/FactFormer` yet making a slight modification for 1D instead of 2D inputs. Li et al. (2023b) deals with 2D inputs in the following manner: given a hidden state of a solution $w$ with with spatial dimensions $S_x$ and $S_y$ and hidden dimension $H$, two queries and keys are built from $w$ by applying two different MLPs ($\text{MLP}_x$ and $\text{MLP}_y$) and then taking the mean across $S_y$ and $S_x$, respectively. Thus, we get $q_x$ and $k_x$ of shape $(S_x, H)$, and $q_y$ and $k_y$ of shape $(S_y, H)$. The attention "values" $v$ of shape $(S_x, S_y, H)$ are obtained from $w$ by a linear layer. Then two linear attention transformations are applied, first with $q_x$ and $k_x$ across the $S_x$ dimension, and then $q_y$ and $k_y$ across the $S_y$ dimension. For our 1D case we do not have $\text{MLP}_y$, nor $q_y, k_y$. Concretely, we only have one $\text{MLP}_x$, we do not take means to compute $q_x$ and $k_x$, and we only apply one linear attention per layer.

**Galerkin Transformer (GKT) (Cao, 2021)**: This model uses four linear attention layers over the spatial sequence length. It includes positional information by concatenating the grid coordinates into the queries, keys and values . After the attention layers, two FNO layers (using all Fourier modes) are used. The hidden dimension used in the experiments is 32 (both for the transformer encoders and the spectral regressor). For the experiments of Figure 1, GKT had unstable performance for some resolutions. Thus, for some resolutions we tried a different training setup: a dropout of 0.05 in the linear attention layer and 0.025 in the FFN layer and 50 training epochs instead of 200. We reported the nRMSE of the best configuration. Specifically, the dropout + reduced training epochs helped performance in resolutions [40-64] for $\nu = 0.075$, [8-60] for $\nu = 0.1$ and [36-48] for $\nu = 0.125$ (all inclusive intervals). The implementation is based on the publicly available code `https://github.com/scaomath/galerkin-transformer`.

**U-Net Neural Operator (U-Net) (Gupta & Brandstetter, 2023)**: This model consists of four down-sample convolution blocks, a middle convolution block, and four upsample convolution blocks. The upsample blocks have residual connections to the downsample blocks in the typical U-Net fashion. The downsample blocks have channel multipliers [1, 2, 2, 2] and no time embeddings are used. The first hidden dimension is 32. The implementation is based on the repository `https://github.com/pdearena/pdearena`.

**S4 - U-Net Neural Operator (S4U-Net)**: This model also uses our MemNO framework. As before, all layers except the memory layer are the same as U-Net. The state dimension is 16 and the we used the S4D kernel. We apply the memory layer after the "middle" convolution block.

### B.1  PARAMETER AND TRAINING TIMES FOR DIFFERENT ARCHITECTURES

The number of parameters of the different baselines and training times (forward + backward) is shown in Table 2.

### B.2  ALGORITHMIC COMPLEXITIES OF S4FFNO AND FFNO

We present the theoretical complexities of the cores of the S4FFNO and FFNO layers (i.e., the spectral convolution of FFNO and the convolution of S4FFNO). Let $S$ be the spatial resolution, $T$

| Architecture | # Params (millions) | Training time (miliseconds) |
|---|---|---|
| Factformer (1D) | 0.65 | 102 |
| GKT | 0.29 | 21 |
| U-Net | 2.68 | 23 |
| FFNO | 4.89 | 28 |
| Multi Input FFNO | 4.89 | 28 |
| S4FFNO | 4.94 | 32 |
| S4U-Net | 2.82 | 25 |

Table 2: Number of parameter and training times (forward and backward pass) of architectures for the experiments in Section 6.1 and Appendix G.3. The batch size is 32, the spatial resolution is 64 and the number of timesteps is 25. The GPU is an NVIDIA L40S.

the number of timesteps, $H$ the hidden dimension and $N$ the state dimension of S4. The core spectral convolution of FFNO (Eq. 16) has a Discrete Fourier Transform across the space dimension and a matrix multiplication in the frequency space, which have complexities $O(TH\tilde{S})$ and $O(TSH^2)$ respectively (tildes denote log factors). In contrast, the S4 layer has a Discrete Fourier Transform across the time dimension and it requires building the convolution kernel, which have complexities $O(SH\tilde{T})$ and $O(SH(\tilde{N}+\tilde{T}))$ respectively (Gu et al., 2022). In our cases, $S$ ranges from 32 to 128, and $T$ is either 20, 25 or 32. Thus, in most cases $O(TH\tilde{T}) < O(TH\tilde{S})$. As for the other term, we use $H = 128$ and $N = 64$, so $N + T \le H$ and we also have $O(SH(\tilde{N}+\tilde{T}))) < O(TSH^2)$. Thus, the S4 memory layer requires less computation than a spatial FFNO layer.

## C  BURGERS' EQUATION (1D): A STUDY ON LOW-RESOLUTION

The Burgers' equation with viscosity $\nu \in \mathbb{R}_+$ is a nonlinear PDE used as a modeling tool in fluid mechanics, traffic flow, and shock waves analysis. It encapsulates both diffusion and advection processes, making it essential for studying wave propagation and other dynamic phenomena. It is known for exhibiting a rich variety of behaviors, including the formation of shock waves and the transition from laminar to turbulent flow. The viscous Burgers' equation is written as:

$$u_t + uu_x = \nu u_{xx}$$

We used the publicly available dataset of the Burgers' equation in the PDEBench repository (Takamoto et al., 2023) with viscosity 0.001, which is available at resolution 1024.

We perform experiments at resolutions 64, 128, 256, 512 and 1024 and show results for the models Galerkin Transformer (GKT) (Cao, 2021), U-Net neural operator (**U-Net**) (Gupta & Brandstetter, 2023), Factorized Fourier Neural Operator (FFNO), Multi Input Factorized Fourier Neural Operator (Multi input FFNO) and our proposed model S4 Factorized Fourier Neural Operator (S4FFNO). The results are shown in Figure 4a.

In low resolutions, memory-based architectures (Multi Input FFNO and S4FFNO) outperform the best Markovian baseline (FFNO). Specifically, S4FFNO achieves more than $4\times$ less error than FFNO in resolutions 32 and 64 (see Table 1). Additionally, S4FFNO has slightly better performance than Multi Input FFNO in high resolutions (512, 1024). Furthermore, we show the difference in nRMSE between FFNO and S4FFNO at each timestep in figure 4b. We observe that at the first timestep there is no difference between the two models—which is expected because S4FFNO has the exact same architecture as FFNO for the first timestep. Yet as the initial condition is rolled out, there is more history of the trajectory and the difference between FFNO and S4FFNO increases.

### C.1  CORRELATION WITH FRACTION OF UNOBSERVED MODES

As mentioned in Section 6.3 we measure the correlation of $\omega_f$ defined in Equation 14 with the difference in the nRMSE between FFNO and S4FFNO. The results can be seen in Figure 5.

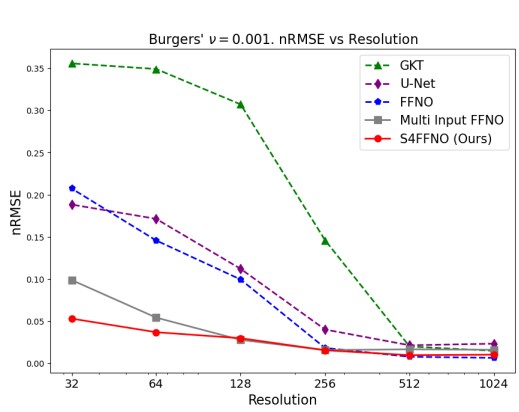

(a) nRMSE of several models at different resolutions.

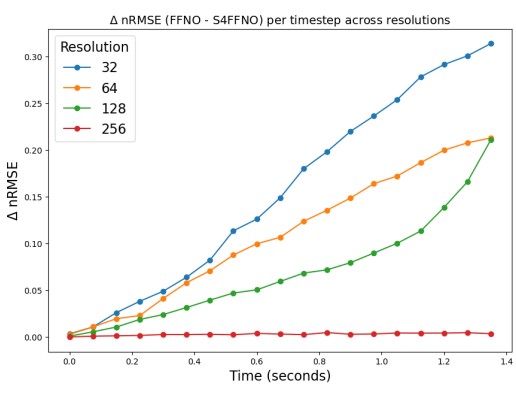

(b) Difference between the nRMSE of FFNO and S4FFNO per timestep (higher difference means better performance of S4FFNO).

Figure 4: Results for the Burgers's PDEBench dataset with viscosity $\nu = 0.001$.

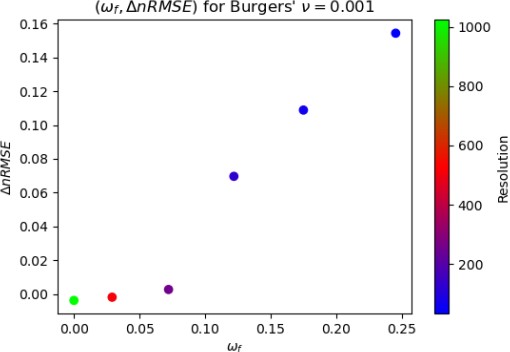

Figure 5: Difference in nRMSE between FFNO and S4FFNO against $\omega_f$ (defined in Equation 14) for different resolutions of the Burgers' Equation. $\omega_f$ is averaged over all trajectories in the dataset and across all timesteps of the experiment. The value is computed approximating the continuous Fourier modes with the Discrete Fourier modes of the solution in the highest resolution available (1024 for Burgers' Equation).

## D    ANALYSIS OF HIGH-FREQUENCY FOURIER MODES IN COMMON 1-D DATASETS

In Section 5.1, we explained that one of the main criteria for choosing the datasets of our experiments was the high contribution from high-frequency Fourier modes in the solutions of the PDEs. Intuitively, when the solution contains contributions from high-frequency Fourier modes, say higher than a number $k$, then it cannot be approximated accurately from its first $k$ Fourier components (see Definition 5). Therefore, when observed at a finite resolution $f$, only $\lfloor \frac{f}{2} \rfloor$ Fourier modes can be estimated[6], which is not enough to approximate the solution when $k \gg \lfloor \frac{f}{2} \rfloor$. In this case, there is an "unobserved" part of the solution (which corresponds to the high-frequency components), and thus we can expect the memory term of the Mori-Zwanzig Equation 6 to be non-negligible.

In order to quantitatively measure the importance of the high-frequency components of a function, we propose using $\omega_f$ from Equation 14. This quantity measures the fraction of Fourier modes (weighted by their amplitude) that are above the frequency $\lfloor \frac{f}{2} \rfloor$, and thus $0 \leq \omega_f \leq 1$. When $\omega_f$ is close to 0, then we expect the solution to be very accurately approximated from the Fourier modes that are observed at resolution $f$, so the "unobserved" part of the function is very small and thus the

---

[6]This is a consequence of Nyquist–Shannon Theorem (Shannon, 1949)

memory term of the Mori-Zwanzig Equation is expected to be negligible. Conversely, if $\omega_f$ is large, we expect the memory term to be significant[7].

The results for $\omega_f$ are shown in Figure 6. For most PDEBench datasets the values of $\omega_f$ are very small, even for very small resolutions like 16 (note that the original data is in resolution 1024). Therefore, based on our previous discussion we expect the memory term to be negligible. On several exploratory experiments on these datasets, we indeed saw no benefit of using memory to model PDEs. The only exception is the Burgers' dataset with viscosity $\nu = 0.001$, where our experiments in Appendix C show a superior performance of memory-augmented models over Markovian ones for resolutions 64, 128 and 256 (Figure 4a).

In the case of the Kuramoto–Sivashinsky (KS) dataset, we again see that the viscosities that are typical in other works, like $\nu = 1.0$ and $\nu = 0.5$ in PDE-Refiner (Lippe et al., 2023), do not have a large $\omega_f$, unless the resolutions are low (16 or 32). For that reason, we generated our own datasets with lower viscosities, which yield higher values for $\omega_f$ and thus a more challenging benchmark to compare Markovian and memory models. Besides the change of viscosity, PDE-Refiner generation method had a warm-up of $T = 72$ seconds, while we did not consider a warm-up. This warm-up explains the higher presence of high frequencies for viscosity $\nu = 0.5$ compared to our viscosities at resolution 16. Details and code to generate our datasets are provided in Appendix E.

It can be seen that $\omega_f$ depends on both the parameters of the PDE (i.e. viscosities in Burgers' and KS) and the observation resolution $f$. Thus, a key to understanding the importance of the memory term is observing the *resolution relative to the Fourier frequency spectrum of the solution*, as we noted in Section 6.1. Additionally, another important characteristic that affects $\omega_f$ is the *frequency spectrum of the initial condition*. While the initial condition for KS is generated as a superposition of sinusoidal waves, PDEBench also uses this superposition of waves but applies some transformations to it, like taking the absolute value (see Appendix D of Takamoto et al. (2023)). These transformations lead to the appearance of higher order frequencies in the initial conditional and thus also affect the frequency spectrum of later timesteps. We believe this is why Burgers' with $\nu = 0.001$ also exhibits high $\omega_f$ at resolutions 128 and 256 (Figure 6).

We hope $\omega_f$ can serve as a practical quantity to help practitioners and researchers explore whether to consider memory architectures or not.

# E  DATA GENERATION

## E.1  KURAMOTO–SIVASHINSKY EQUATION

The Kuramoto-Sivashinsky (KS) equation is given by:

$$u_t + uu_x + u_{xx} + \nu u_{xxxx} = 0 \quad (t, x) \in [0, T] \times [0, L]$$
$$u(0, x) = u_0(x) \quad x \in [0, L]$$

We use periodic boundary conditions. Our data generation method is very similar to the one used in PDERefiner (Lippe et al., 2023), except for the three following differences: (1) We do not have a random $\Delta t$ per trajectory (2) We set the initial condition to have eight Fourier modes in the spectrum, whereas Lippe et al. (2023) uses three (3) We do not discard the first generated timesteps of the solution of the PDE. We provide a forked repository with these changes in https://github.com/r-buitrago/LPSDA, which is based on the original repository of Brandstetter et al. (2022) https://github.com/brandstetter-johannes/LPSDA. The generation command for our datasets is (change `--viscosity` for the desired value):

```
python generate/generate_data.py --experiment=KS --train_samples=2048
--valid_samples=256 --test_samples=0 --L=64 --nt=51 --nx=512
--nt_effective=51 --viscosity=0.1 --end_time=5.0 --lmax=8
```

Now, we give an explanation of the generation procedure. We employ the *method of lines* (Schiesser, 1991), where the spatial dimension is discretized, and the PDE is transformed to a system of Or-

---

[7]This is not a precise mathematical argument, but rather an intuition that has proven to be helpful in practice for the PDEs we have considered. In general, the interaction of the PDE and the frequency spectrum of the solution is complex and $\omega_f$ by itself is not enough to determine the magnitude of the memory term.

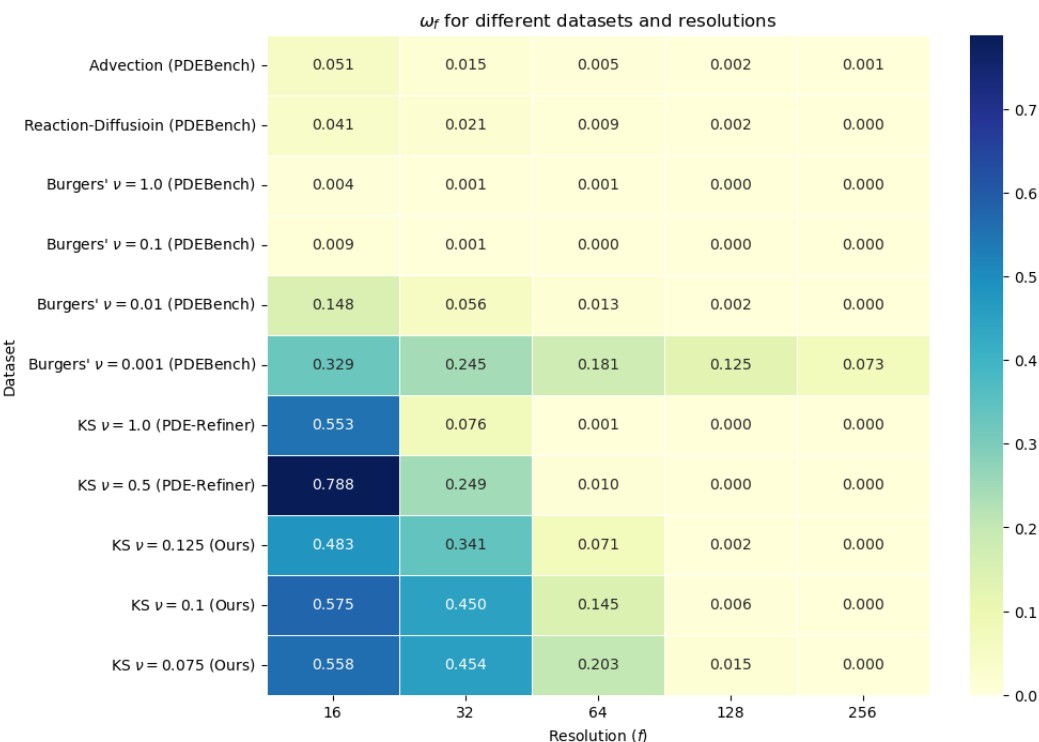

Figure 6: $\omega_f$ for different resolutions $f$ and datasets. $\omega_f$ measures the ratio of Fourier modes that are above frequency $\frac{f}{2}$ (see Equation 14). The Advection, Diffusion-Reaction and Burgers' datasets come from PDEBench (Takamoto et al., 2023) (the Diffusion-Sorption dataset is not considered because it does not have periodic boundary conditions). The KS datasets come from either PDE-Refiner (Lippe et al., 2023), or they are generated by ourselves following Section E. The PDE-Refiner datasets use a $T_{\text{warm-up}} = 72s$, that is, they discard all timesteps of the numerical solvers up to time $72s$. In contrast, our generated KS datasets do not have warm-up period. $\omega_f$ is averaged across all trajectories in the dataset, and also averaged across the first 20 timesteps. The values of $\omega_f$ are computed approximating the continuous Fourier modes with Discrete Fourier modes of the solution in the highest resolution available (512 for KS datasets and 1024 for all other PDEBench datasets).

dinary Differential Equations (ODEs), one per point in the grid. In order to compute the spatial derivative of the solution at each point in the grid, a pseudospectral method is used, where derivatives are computed in frequency space and then converted to the original space through a Fast Fourier Transform. This method is implemented in the `diff` method of the `scipy.fftpack` package (Virtanen et al., 2020). Similarly, the system of ODEs is solved numerically with a implicit Runge-Kutta method of the Radau IIA family of order 5 (Hairer & Wanner, 1996), which is implemented in the `solve_ivp` method of `scipy.integrate`. We refer to the code provided in Brandstetter et al. (2022) to reproduce this data generation, however certain small modifications have to be made, like using a fixed $\Delta t$ per trajectory and increasing the number of modes in the initial condition.

As for the PDE parameters, we use $L = 64$ and $T = 2.5$. For the initial condition, we use a superposition of sinusoidal waves:

$$u_0(x) = \sum_{i=0}^{20} A_i \sin\left(\frac{2\pi k_i}{L}x + \phi_i\right)$$

where for each trajectory, the $A_i$ are sampled from a continuous uniform in $[-0.5, 0.5]$, the $k_i$ are sampled from a discrete uniform in $\{1, 2, ..., 8\}$, and the $\phi_i$ are sampled from a uniform uniform in $[0, 2\pi]$. We discretize $[0, T]$ into 26 equispaced points separated by $\Delta t = 0.1$. In the experiments in Section 6.1, for each of the four values of the viscosity $(0.15, 0.125, 0.1, 0.075)$, we generated a dataset with spatial resolution 512 with 2048 training samples and 256 test samples. For the experiment in the sequential model ablation in section G.1, we generated one dataset with viscosity 0.15 in resolution 256, 4096 training samples and 256 test samples.

### E.2 BURGERS' 1D EQUATION

The 1D Burgers' equation can be written as:

$$u_t + uu_x = \nu u_{xx} \quad (t, x) \in [0, T] \times [0, L]$$

For the Burgers' equation, we take the publicly available Burgers' dataset of PDEBench (Takamoto et al., 2023) with viscosity 0.001. Out of the 10000 samples of the dataset, we use $10\%$ for testing. For training, we found it sufficient to use 2048 samples. Additionally, for training and testing we only used the 20 first timesteps, since we observed that after the 20th timestep the diffusion term of the equation $u_{xx}$ attenuates all high frequencies and the solution changes very slowly.

### E.3 NAVIER-STOKES 2D EQUATION

The incompressible Navier-Stokes equation in the 2D unit torus is given by:

$$\frac{\partial w(x, t)}{\partial t} + u(x, t) \cdot \nabla w(x, t) = \nu \Delta w(x, t) + f(x), \qquad x \in (0, 1)^2, t \in (0, T]$$

$$\nabla \cdot u(x, t) = 0, \qquad x \in (0, 1)^2, t \in [0, T]$$

$$w(x, 0) = w_0(x), \qquad x \in (0, 1)^2$$

For the data generation, we follow the method of Li et al. (2021), yet with different temporal and spatial grids. The initial conditions $w_0$ are sampled from a Gaussian Random field $\mathcal{N}\left(0, 7^{\frac{3}{2}}(-\Delta + 49I)^{-2.5}\right)$ with periodic boundary conditions. The forcing term is $f(x_1, x_2) = 0.1\left(\sin 2\pi(x_1 + x_2) + \cos 2\pi(x_1 + x_2)\right)$. At each timestep, the velocity is obtained from the vorticity by solving a Poisson equation. Then, spatial derivatives are obtained, and the non-linear term is computed in the physical space and then dealiased. A Crank-Nicholson scheme is used to move forward in time, with a timestep of $10^{-4}$. We use a 512x512 spatial grid which is then downsampled to 64x64 for our experiments. For the viscosity $\nu = 10^{-3}$, we use a final time of 16 seconds and sample every 0.5 seconds. For the viscosity $\nu = 10^{-5}$, we use a final time of 3.2 seconds and sample every 0.1 seconds. For more details on the data generation algorithm, we refer to Li et al. (2021).

## F TRAINING DETAILS

In this section, we will provide a detailed description of the training hyperparameters used in the KS experiments of Section 6.1, in the Burgers experimente of section C and the Navier Stokes

experiments of section 6.2. We start with the training hyperparameters. All our experiments used a learning rate of $0.001$. For the number of epochs, in KS and Burgers, the training was done over 200 epochs with cosine annealing learning scheduling (Loshchilov & Hutter, 2017); whereas in Navier Stokes we trained for 300 epochs and halved the learning rate every 90. As for the number of samples, KS and Burgers were trained with 2048 samples and Navier Stokes with 1024 samples. Lastly, we observed that the batch size was a sensitive hyperparameter for both the memory and memoryless models (it seemed to affect both equally) so we ran a sweep at each experiment to select the best performing one. In the results shown in the paper, KS and Navier Stokes use a batch size of 32, and Burgers a batch size of 64.

Another relevant detail is the memory length in training, that is, the number of past states that were fed to the memory layer in the MemNO model. In the KS and Burgers experiments, the maximum memory lengths are 20 and 25 (which are the same as the number of timesteps of the dataset). That means that for the last timestep, the previous 19 or 24 states were fed into the memory layer. However, for GPU memory limitations in Navier Stokes the memory length was 16, half the number of timesteps of each trajectory in the dataset.[8] In this case, the memory was reset after the 16th timestep, i.e. for the 16th timestep the 15 past states were fed to the memory model, yet for the 17th timestep only the 16th timestep was fed. Then, for the 18th timestep, the 17th and 16th were fed, and so on.

As in (Tran et al., 2023), experiments were trained using teacher forcing. This means that for the prediction of the $i$-th timestep during training, the ground truth of the $i-1$ previous steps was fed to the model (as opposed to the prediction of the model for such steps).

We ran our experiments on A6000/A6000-Ada GPUs. The Navier Stokes 2D experiments required around 34GB of GPU memory for the batch size of 32 and took around 5 hours to finish, whereas the rest of experiments in 1D required a lower GPU memory (less than 10GB) and each run took around 1 or 2 hours, depending on the resolution.

# G    ABLATIONS ON THE MEMNO ARCHITECTURE

In this section we present three ablations regarding the MemNO architecture

## G.1    ABLATION: CHOICE OF SEQUENTIAL MODEL

In section 5.3 we introduced MemNO as an architecture framework which allowed the introduction of memory through any choice of a sequential layer, which we chose as S4 in the previous experiments. In this section, we explore two other candidates for the sequential layers: a Transformer and an LSTM. We introduce **Transformer-FFNO (T-FFNO)** and **LSTM-FFNO** as two models that are identical to S4FFNO except in the sequential layer, where a Transformer and an LSTM are used respectively. The Transformer layer includes causal masking and a positional encoding, which is defined for $pos$ across the time dimension and $i$ across the hidden dimension by:

$$PE(pos, 2i) = \sin\left(\frac{pos}{10000^{\frac{2i}{\text{dim\_model}}}}\right)$$
$$PE(pos, 2i+1) = \cos\left(\frac{pos}{10000^{\frac{2i}{\text{dim\_model}}}}\right)$$

We show results for the KS dataset with viscosity $\nu = 0.15$ and different resolutions. This dataset was generated using a resolution of 256 and contains 4096 samples, twice as many compared to the KS datasets of E, given that Transformers are known to perform better in high-data regimes. The results are shown in Figure 7. TFFNO performs significantly worse than S4FFNO across almost all resolutions, and even performs worse than FFNO. In contrast, LSTM-FFNO outperforms FFNO, which shows that MemNO can work with other sequential models apart from S4. The memory term in Equation 6 is a convolution in time, which is equivalent to the S4 layer and very similar to a

---

[8]Under this setup, the GPU memory requirements were around 34 GB. Using the full 32 timesteps for training would require a memory beyond 48GB, which was beyond our GPU capacity (A6000/A6000-Ada GPUs).

Recurrent Neural Network (RNN) style layer, as showed in Gu et al. (2022). We believe that this inductive bias in the memory layer is the reason why both S4FFNO and LSTM-FFNO outperform FFNO. However, S4 was designed with a bias for continuous signals and has empirically proven better performance in these kind of tasks (Gu et al., 2022), which is in agreement with its increased performance over LSTMs in this experiment. Additionally, we observed that LSTMs were unstable to train in Navier Stokes 2D datasets.

Lastly, we make two remarks. Firstly, we believe that Transformers performed worse due to over-fitting, given that the train losses were normally comparable or even smaller than the train losses of the rest of the models at each resolution. We hypothesize that the full access to the past of Transformers models might lead to exploiting spurious correlations during training. Modifications of the Transformer layer or to the training hyperparameters as in other works (Hao et al., 2024; Cao, 2021; Hao et al., 2023) might solve this issue. Secondly, recently there has been a surge of new sequential models such as Mamba (Gu & Dao, 2023; Dao & Gu, 2024), RWQK (Peng et al., 2023), xLSTM (Beck et al., 2024) or LRU (Orvieto et al., 2023). We chose S4 over Mamba-type architectures because in our experiments the PDE temporal dynamics do not change, and thus we do not expect the input-dependent selectivity mechanism to be necessary. However, we leave it as future work to study which of these sequential model has better overall performance, and hope that our study on the settings where the memory effect is relevant can help make accurate comparisons between them.

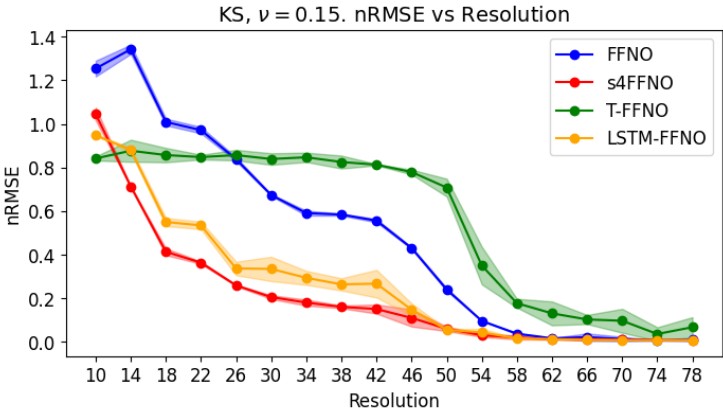

Figure 7: Performance of FFNO, S4FFNO and T-FFNO and LSTM-FFNO in KS with viscosity $\nu = 0.15$.

### G.2  ABLATION: MEMORY LAYER CONFIGURATION

In Section 5.3 we introduced the memory layer in MemNO as a single layer to be interleaved with neural operator layers. In our experiments, we inserted it after the second layer of a four layer neural operator. In this section, we explore the impact of having different layer configurations, including the possibility of having several memory layers. We will denote the configurations with a sequence of S and T letters. S means a neural operator layer (some sort of Spatial convolution), and T a memory layer (some sort of Time convolution). For example, *SSTSS* denotes the architecture of our experiments, where we have 2 neural operators layers, followed by a memory layer, followed by other 2 neural operator layers. Similarly, *SSSST* denotes 4 neural operators layers followed by a memory layer. In Table 3, we present the results for the KS dataset with $\nu = 0.1$ and final time of 4 seconds for several models. We include the S4FFNO model we used in previous experiments in the first row (with configuration *SSTSS*), and the FFNO model in the last row. In the middle rows, we show different configurations of memory and neural operator layers. It can be observed that all models with at least a memory layer outperform FFNO. There are slight differences between configurations, yet we focused mainly on the comparison to the memoryless model. For that reason, we fixed *SSTSS* configuration in our previous experiment, which was the most efficient (only one memory layer) and symmetric. We leave as further work determining if there are settings where a given configuration pattern can be substantially better than the rest.

| Architecture | nRMSE ↓ | | |
|---|---|---|---|
| | Resolution 32 | Resolution 48 | Resolution 64 |
| S4FFNO (*SSTSS*) | $0.123 \pm 0.011$ | $0.086 \pm 0.004$ | $\mathbf{0.015 \pm 0.001}$ |
| S4FFNO (*SSSST*) | $0.142 \pm 0.009$ | $0.069 \pm 0.001$ | $0.017 \pm 0.001$ |
| S4FFNO (*STSSTS*) | $0.141 \pm 0.006$ | $\mathbf{0.064 \pm 0.002}$ | $0.019 \pm 0.001$ |
| S4FFNO (*STSTSTST*) | $\mathbf{0.113 \pm 0.006}$ | $0.070 \pm 0.004$ | $0.017 \pm 0.001$ |
| S4FFNO (*TSSSS*) | $0.129 \pm 0.007$ | $0.080 \pm 0.003$ | $0.017 \pm 0.001$ |
| FFNO | $0.294 \pm 0.004$ | $0.138 \pm 0.013$ | $0.021 \pm 0.002$ |

Table 3: KS, $\nu = 0.1$. The final time is 4 seconds and the trajectories contain 20 timesteps. For each architecture, we tried 4 learning rates (0.002, 0.001, 0.0005 and 0.00025, each with three different seeds. We present the results of the learning rate with the lowest nRMSE averaged across the three seeds. The standard deviation is also with respect to the seeds.

### G.3 ABLATION: S4U-NET

The experiments in Section 6 used S4FFNO as our proposed memory model, which was the instantiation of the MemNO framework (Section 5.3) with FFNO as the Neural Operator and S4 as the memory layer. In the ablations of Section G.1 we showed that although S4 was the best performing memory model, LSTM also provided good performance, showing the versatility of the framework and the importance of adding memory (regardless of the specific architecture). In this section, we show that the MemNO can also use a different Neural Operators as the Markovian layer. In particular, we instantiate the MemNO framework using U-Net as the Markovian Neural Operator, and S4 as the memory layer with a state dimension of 16.

The results for the KS experiment (Section 6.1) and Burgers' experiment (section C) are shown in Table 4. As in the case of S4FFNO, S4U-Net also improves the performance of U-Net in the cases of low resolution. This shows that modeling memory is useful in low resolution for architectures other than FFNO. However, S4U-Net has worse performance than S4FFNO. We reiterate that the main contribution of our work is studying *when* modeling memory is helpful, as well as providing flexible ways to incorporate it into existing neural operators. We leave for future work the study of the trade-offs between different memory models under different setups.

| Architecture | Uses memory | Resolution | nRMSE ↓ | | | |
|---|---|---|---|---|---|---|
| | | | KS | | | Burgers' |
| | | | $\nu = 0.075$ | $\nu = 0.1$ | $\nu = 0.125$ | $\nu = 0.001$ |
| U-Net | No | 32 | 0.542 | 0.511 | 0.249 | 0.188 |
| S4U-Net (Ours) | Yes | | **0.364** | **0.277** | **0.104** | **0.096** |
| U-Net | No | 64 | 0.147 | 0.062 | **0.022** | 0.171 |
| S4U-Net (Ours) | Yes | | **0.114** | **0.052** | 0.026 | **0.070** |
| U-Net | No | 128 | **0.033** | **0.027** | **0.014** | 0.112 |
| S4U-Net (Ours) | Yes | | 0.058 | 0.030 | 0.022 | **0.057** |

Table 4: nRMSE values for the S4U-Net architecture at different resolutions for Burgers' and KS with different viscosities. The values of U-Net are the same as the ones in Table 1 and are provided here for context. More details on training are given in Appendix F, on the KS experiment on 6.1 and on the Burgers' experiment in Appendix C.

## H ABLATIONS ON S4FFNO AND FFNO PERFORMANCE

### H.1 ABLATION: MEMORY WINDOW LENGTH

In this section we present an ablation on the memory window length of the S4FFNO architecture. We recall that in our experiments of Section 6.1 we had a discretized grid $\mathcal{T} = [t_0, t_1, ..., t_N]$, and

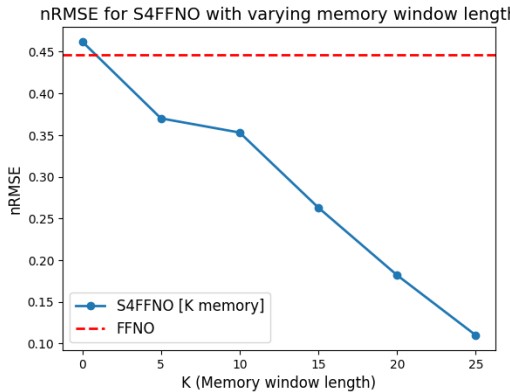

Figure 8: nRMSE for S4FFNO with varying memory window length, for the KS experiment of Section 6.1 ($\nu = 0.1$ and resolution 32). A memory window of $K$ means that the S4 model only has access to the memory of the last $K$ timesteps to predict the next one. At training time, the sequence length is split into chunks of $K$ timesteps and each chunk is trained independently. At inference time, the S4FFNO is given access to the last K predicted timesteps to make the next prediction.

in order to predict the solution at timestep $t_i$ S4FFNO had access to the memory of all previous timesteps (i.e. the S4 model operated on the hidden dimensions $v(t_j)$ for $0 \leq j \leq i - 1$). In this section, we study what happens when S4 is only fed the last $K$ timesteps, i.e. in order to predict $t_i$, S4 operates on the solution at timesteps $t_{i-K}, t_{i-K+1}, ..., t_i$. The results are shown in Figure 8.

S4FFNO improves performance as the window length increases, illustrating that the reason for the increased performance of S4FFNO is the capacity to model the memory of past timesteps.

## H.2 Ablation: FFNO model size

Now we consider what happens when we increase the model size of FFNO. Based on the results of Section 6.1, S4FFNO outperformed FFNO when they had similar compute budgets (see Table 2). However, S4FFNO still outperforms FFNO when FFNO has a much higer compute and parameter budget, as it can be seen in Table 5. Thus, we conclude that S4FFNO has superior performance due to the possibility of modeling memory from past states.

| Architecture | Hidden Dimension | # Layers | nRMSE ↓ | | # Params (millions) | |
|---|---|---|---|---|---|---|
| | | | Resolution 32 | Resolution 48 | Resolution 32 | Resolution 48 |
| S4FFNO | 128 | 4 | **0.108** | **0.045** | 2.8 | 3.9 |
| FFNO | 128 | 4 | 0.440 | 0.238 | 2.8 | 3.8 |
| FFNO | 128 | 8 | 0.361 | 0.181 | 5.5 | 7.6 |
| FFNO | 256 | 4 | 0.435 | 0.252 | 11.1 | 15.3 |
| FFNO | 256 | 8 | 0.346 | 0.194 | 22.2 | 30.6 |

Table 5: Performance of S4FFNO and different model sizes of FFNO on the KS experiment with viscosity $\nu = 0.1$ and resolutions 32 and 48. The experimental details are the same as 6.1.

## I Ablation on the modeling of local spatial information

In Section 5.3 we presented the MemNO framework as a way to build neural operators that model memory, which is inspired by the Mori-Zwanzig formalism. In Equation 6, it can be seen that the memory term depends on the differential operator $\mathcal{L}$, which can depend on the spatial derivatives of the input. The memory layer of MemNO is applied independently to each spatial dimension of the hidden representation of the input, and thus it does not model spatial derivatives explicitly, although the hidden dimension can in principle encode such local information implicitly.

In this section, we present several modifications to the S4FFNO architecture, which attempt to provide an inductive bias for modeling spatial information more explicitly. In particular, we try the following modifications:

- **S4FFNO + Input Gradients**: The numerical gradients of the input $u_t$ are fed to the model encoder. Specifically, $u_t$ is represented as a vector of spatial shape $S$, and the spatial gradients are approximated using second-order accurate central differences method (using the method `torch.gradient` of the `pytorch` library (Paszke et al., 2019)). This would directly facilitate storing information about the first order spatial gradients in the hidden dimension.
- **S4FFNO + Convolutional Encoder**: Instead of having a linear layer as encoder, we substitute it with a 1D Convolution of kernel size 3 across the spatial dimension.
- **S4FFNO + Convolution before memory layer**: Before feeding the sequence of hidden presentations $[v_0, v_1, ..., v_t]$ to the memory layer S4, we apply a 1D Convolution across the spatial dimension with kernel size of 3.

The performance of such modifications on the KS experiment of Section 6.1 with viscosity $\nu = 0.1$ is shown in Table 6. It can be seen that none of the modifications improve performance compared to the baseline significantly. We hypothesize that training becomes more difficult with these architecture modifications, and thus they do not provide an improvement in performance. We leave as future work to consider other architectures to model spatial information more explicitly.

| Architecture | nRMSE ↓ | | | | |
|---|---|---|---|---|---|
| | Resolution 16 | Resolution 32 | Resolution 48 | Resolution 64 | Resolution 80 |
| FFFNO | 1.106 | 0.4461 | 0.2325 | 0.328 | **0.0040** |
| S4FFNO (Base) | **0.3318** | 0.1081 | **0.0455** | **0.0111** | 0.0048 |
| S4FFNO + Input Gradients | 0.4433 | 0.1120 | 0.0482 | 0.0120 | 0.0054 |
| S4FFNO + Convolutional encoder | 0.4340 | **0.1053** | 0.0571 | 0.0138 | 0.0065 |
| S4FFNO + Convolution before memory layer | 0.4283 | 0.1224 | 0.0491 | 0.0142 | 0.0057 |

Table 6: Performance comparison between FFNO, S4FFNO and several architecture modifications to S4FFNO aimed at introducing an inductive bias to model spatial information, see Section I. The experiment is performed on the the KS PDE with viscosity $\nu = 0.1$ under the same setup as Section 6.1. The architecture details of S4FFNO and FFNO are provided in Appendix B.

## J  MEMNO DIAGRAM

In Figure 9 we provide a diagram for the MemNO framework (Section 5.3) in 1D.

## K  QUANTIFYING THE EFFECT OF MEMORY

We include the proof for Theorem 1.

*Proof.* We proceed to the Equation 9 first. Note that $u_1(t), \forall t \geq 0$ can be written as $u_1(t) = a_0^{(t)} \mathbf{e}_0 + a_1^{(t)} \mathbf{e}_1$. Moreover, by Proposition 1, we have

$$\frac{\partial a_0^{(t)}}{\partial t} = 2B a_1^{(t)} \tag{17}$$

$$\frac{\partial a_1^{(t)}}{\partial t} = a_1^{(t)} + B a_0^{(t)} \tag{18}$$

In matrix form, these equations form a linear matrix ODE:

$$\frac{\partial}{\partial t} \begin{pmatrix} a_0^{(t)} \\ a_1^{(t)} \end{pmatrix} = \begin{pmatrix} 0 & 2B \\ B & 1 \end{pmatrix} \begin{pmatrix} a_0^{(t)} \\ a_1^{(t)} \end{pmatrix}$$

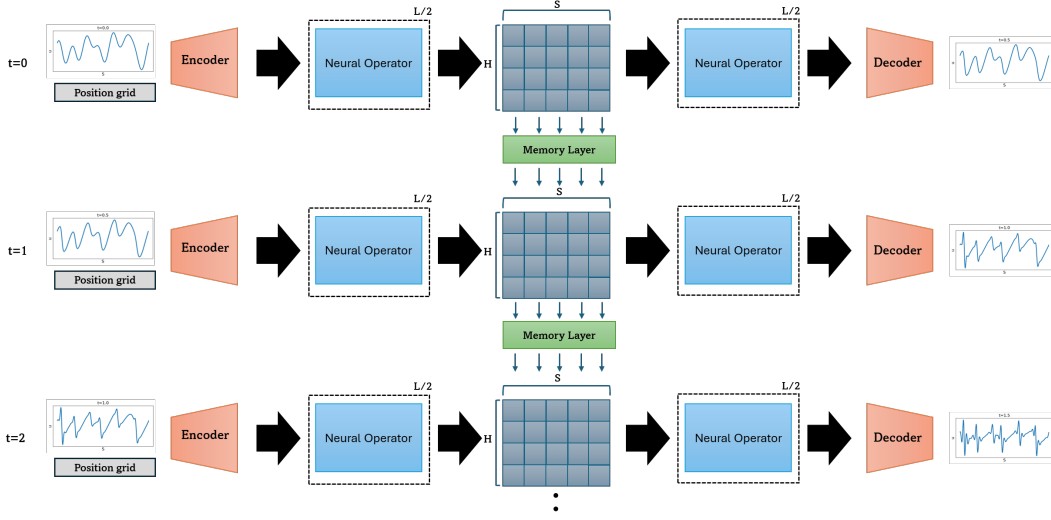

Figure 9: Diagram of the MemNO framework in 1D (Section 5.3). $S$ denotes spatial dimension, $H$ denotes hidden dimension, and $L$ number of layers. The memory layer in inserted in the middle, although the framework works with other configurations.

The solution of this ODE is given by $\begin{pmatrix} a_0^{(t)} \\ a_1^{(t)} \end{pmatrix} = \exp\left( t \begin{pmatrix} 0 & 2B \\ B & 1 \end{pmatrix} \right) \begin{pmatrix} a_0^{(0)} \\ a_1^{(0)} \end{pmatrix}$. By the first statement of Lemma 1 and the non-negativity of $a_0^{(0)}, a_1^{(0)}$, we get:

$$a_0^{(t)} \leq 10 e^{\sqrt{2}Bt} \left( a_0^{(0)} + a_1^{(0)} \right), \tag{19}$$

$$a_1^{(t)} \leq 10 e^{\sqrt{2}Bt} \left( a_0^{(0)} + a_1^{(0)} \right) \tag{20}$$

We proceed to Equation 10. Note that for any $s \geq 0$, we can write $u_2(s) = \hat{a}_0^{(s)} \mathbf{e}_0 + \hat{a}_1^{(s)} \mathbf{e}_1$ with $\hat{a}_0^{(0)} = a_0^{(0)}$ and $\hat{a}_1^{(0)} = a_1^{(0)}$. By Proposition 1, we have

$$\mathcal{QL}u_2(x) = B\hat{a}_1^{(s)} \mathbf{e}_2(x)$$

Moreover, given a function $v(x)$, the action of the operator $\exp\{\mathcal{QL}(\tilde{t})\}$ on $v$ is given by the solution $w(\tilde{t}, x)$ to the PDE

$$\frac{\partial}{\partial t} w(t, x) = \mathcal{QL}w(t, x)$$

$$w(0, x) = v(x)$$

If $w(t, x) = \sum_{n \in \mathbb{N}_0} b_n^{(t)} \mathbf{e}_n$ and $\forall n \in \mathbb{N}_0, b_n^{(0)} \geq 0$, we are interested in solving the previous PDE with initial conditions $b_2^{(0)} = B\hat{a}_1^{(s)}$ and $b_n^{(0)} = 0 \ \forall n \neq 2$.

We claim that the coefficients $\hat{a}_n^{(t)} \geq 0 \ \forall t > 0$ and $\forall n \in \{0, 1\}$. For $t = 0$ this is by definition, and we will prove it for all $t$ by way of contradiction. Suppose the claim is not true, then there exists a $t^* > 0$, and some $n^* \in \{0, 1\}$ such that $\hat{a}_{n^*}^{(t^*)} = 0$, and $\hat{a}_n^{(s)} > 0 \ \forall n \in \{0, 1\}$ and $\forall s < t^*$. But from continuity this implies that there exists $0 < t' < t^*$ such that $\frac{\partial}{\partial t} \hat{a}_{n^*}^{(t')} < 0$. However, it can be easy to see that if $\hat{a}_n^{(s)} > 0 \ \forall s \leq t'$, then $\mathcal{P}_1 \mathcal{L} u_2(t') > 0$ and $\mathcal{P}_1 \mathcal{L} \int_0^{t'} \exp\{\mathcal{QL}(t-s)\} u_2(s) ds > 0$. Therefore, from Equation 10, $\frac{\partial}{\partial t} \hat{a}_{n^*}^{(t')} > 0$, which is a contradiction.

This claim implies that $b_n^{(0)} \geq 0 \ \forall n \in \mathbb{N}$, and in turn it implies that $b_n^{(t)} \geq 0 \ \forall n \in \mathbb{N}, t > 0$. Applying $\mathcal{QL}$ results in the following inequalities for the coefficients $b_1^{(t)}, b_2^{(t)}, b_3^{(t)}$:

$$\frac{\partial}{\partial t} b_1^{(t)} \geq b_1^{(t)} + B b_2^{(t)} \geq B b_2^{(t)} \tag{21}$$

$$\frac{\partial}{\partial t} b_2^{(t)} \geq B b_1^{(t)} + 4 b_2^{(t)} + B b_3^{(t)} \geq B b_1^{(t)} + B b_3^{(t)} \tag{22}$$

$$\frac{\partial}{\partial t} b_3^{(t)} \geq B b_2^{(t)} + 9 b_3^{(t)} \geq B b_2^{(t)} \tag{23}$$

Thus, we can write a linear matrix ODE for the vector $(b_1^{(t)}, b_2^{(t)}, b_3^{(t)})$:

$$\frac{\partial}{\partial t} \begin{pmatrix} b_1^{(t)} \\ b_2^{(t)} \\ b_3^{(t)} \end{pmatrix} \geq \begin{pmatrix} 0 & B & 0 \\ B & 0 & B \\ 0 & B & 0 \end{pmatrix} \begin{pmatrix} b_1^{(t)} \\ b_2^{(t)} \\ b_3^{(t)} \end{pmatrix} \tag{24}$$

Therefore, using Lemma 2, for sufficiently large $B$ we have $b_2^{(t-s)} \geq \frac{B e^{\sqrt{2} B(t-s)}}{10} \hat{a}_1^{(s)}$.

Hence, if we write $\int_0^t \exp\{\mathcal{Q}\mathcal{L}(t-s)\} \mathcal{Q}\mathcal{L} u_2(s) ds$ in the basis $\{\mathbf{e}_n\}_{n \in \mathbb{N}_0}$, the coefficient for $\mathbf{e}_2$ will be lower bounded by

$$\int_0^t \frac{1}{10} B e^{B(t-s)} a_1^{(s)} ds$$

Applying the second statement of Lemma 1 and using the non-negativity of $a_0^{(0)}$ and $a_1^{(0)}$, we have $\hat{a}_1^{(s)} \geq \frac{1}{10} e^{\sqrt{2} B s} \left( a_0^{(0)} + a_1^{(0)} \right)$. Hence, the coefficient for $\mathbf{e}_2$ is lower bounded by

$$\int_0^t \frac{1}{10} B e^{\sqrt{2} B(t-s)} \frac{1}{10} e^{\sqrt{2} B s} \left( a_0^{(0)} + a_1^{(0)} \right) ds \geq \frac{Bt}{100} e^{\sqrt{2} B t} \left( a_0^{(0)} + a_1^{(0)} \right)$$

We finally need to consider what happens after applying the outermost operator $\mathcal{P}_1 \mathcal{L}$. Because of Proposition 1 again, applying $\mathcal{L}$ makes the coefficient in front of $\mathbf{e}_1$ at least $\frac{B^2 t}{100} e^{\sqrt{2} B t} \left( a_0^{(0)} + a_1^{(0)} \right)$. Finally, applying $\mathcal{P}_1$ preserves the coefficient in front of $\mathbf{e}_1$.

Hence, equation Equation 10 results in the following evolution inequalities:

$$\frac{\partial \hat{a}_0^{(t)}}{\partial t} \geq 2 B \hat{a}_1^{(t)} \tag{25}$$

$$\frac{\partial \hat{a}_1^{(t)}}{\partial t} \geq \hat{a}_1^{(t)} + B \hat{a}_0^{(t)} + \frac{B^2 t}{100} e^{\sqrt{2} B t} \left( a_0^{(0)} + a_1^{(0)} \right) \tag{26}$$

Using the second statement of Lemma 1 again we have that $\hat{a}_0(t) \geq \frac{1}{10} e^{\sqrt{2} B s} \left( a_0^{(0)} + a_1^{(0)} \right)$. Thus, dropping the (positive) term $\hat{a}_1^{(t)}$ in equation 26, we have:

$$\frac{\partial \hat{a}_1^{(t)}}{\partial t} \geq \left( \frac{1}{10} + \frac{Bt}{100} \right) B e^{\sqrt{2} B t} \left( a_0^{(0)} + a_1^{(0)} \right) \tag{27}$$

Integrating this equation yields:

$$\hat{a}_1^{(t)} \geq a_1^{(0)} + \frac{1}{200} e^{\sqrt{2} B t} \left( \sqrt{2} B t + 10 \sqrt{2} - 1 \right) \left( a_0^{(0)} + a_1^{(0)} \right) \tag{28}$$

Thus, we have $a_1^{(t)} \gtrsim B t e^{\sqrt{2} B t} \left( a_0^{(0)} + a_1^{(0)} \right)$. Together with equation 19, the claim of the Theorem follows. $\qquad\square$

**Lemma 1.** *There exists $B > 0$ sufficiently large such that for all $t > 0$ the matrix $\begin{pmatrix} 0 & 2Bt \\ Bt & t \end{pmatrix}$ satisfies:*

$$\forall i, j \in \{1, 2\}, \exp\left( \begin{pmatrix} 0 & 2Bt \\ Bt & t \end{pmatrix} \right)_{i,j} \leq 10 \exp\left( \sqrt{2} B t \right) \tag{29}$$

$$\forall i, j \in \{1, 2\}, \exp\left( \begin{pmatrix} 0 & 2Bt \\ Bt & t \end{pmatrix} \right)_{i,j} \geq \frac{1}{10} \exp\left( \sqrt{2} B t \right) \tag{30}$$

*Proof.* By direct calculation, we have:

$$\exp\left(\begin{pmatrix} 0 & 2Bt \\ Bt & t \end{pmatrix}\right) = \frac{1}{2\sqrt{8B^2+1}}\begin{pmatrix} \sqrt{8B^2+1}g(B,t) - h(B,t) & 4Bh(B,t) \\ 2Bh(B,t) & \sqrt{8B^2+1}g(B,t) + h(B,t) \end{pmatrix}$$

where:

$$g(B,t) = e^{\frac{1}{2}(\sqrt{8B^2+1}+1)t} + e^{-\frac{1}{2}(\sqrt{8B^2+1}-1)t}$$
$$h(B,t) = e^{\frac{1}{2}(\sqrt{8B^2+1}+1)t} - e^{-\frac{1}{2}(\sqrt{8B^2+1}-1)t}$$

Thus, the statement follows.

$\square$

**Lemma 2.** *For all $B > 0$, the matrix* $\begin{pmatrix} 0 & B & 0 \\ B & 0 & B \\ 0 & B & 0 \end{pmatrix}$ *satisfies:*

$$\forall i,j \in \{1,2,3\}, \exp\left(\begin{pmatrix} 0 & B & 0 \\ B & 0 & B \\ 0 & B & 0 \end{pmatrix}\right)_{i,j} \geq \frac{1}{10}\exp\left(\sqrt{2}B\right) \tag{31}$$

*Proof.* By direct calculation:

$$\exp\left(\begin{pmatrix} 0 & B & 0 \\ B & 0 & B \\ 0 & B & 0 \end{pmatrix}\right)_{i,j} =$$

$$\frac{1}{4}e^{-\sqrt{2}B}\begin{pmatrix} 2e^{\sqrt{2}B} + e^{2\sqrt{2}B} + 1 & \sqrt{2}e^{2\sqrt{2}B} - \sqrt{2} & -2e^{\sqrt{2}B} + e^{2\sqrt{2}B} + 1 \\ \sqrt{2}e^{2\sqrt{2}B} - \sqrt{2} & 2(e^{2\sqrt{2}B} + 1) & \sqrt{2}e^{2\sqrt{2}B} - \sqrt{2} \\ -2e^{\sqrt{2}B} + e^{2\sqrt{2}B} + 1 & \sqrt{2}e^{2\sqrt{2}B} - \sqrt{2} & 2e^{\sqrt{2}B} + e^{2\sqrt{2}B} + 1 \end{pmatrix}$$

Thus, the statement follows.

$\square$

## L DEFINITION OF PERIODIC BOUNDARY CONDITIONS

For completeness, we give a precise definition of periodic boundary conditions for the PDE defined in Definition 2:

**Definition 6** (Periodic Boundary Conditions). For a PDE given by Definition 2 with $\Omega = [0,L]^d$, we define the periodic boundary conditions as the condition:

$$u(x_1, \cdots, x_{k-1}, 0, x_{k+1}, \cdots x_d) = u(x_1, \cdots, x_{k-1}, L, x_{k+1}, \cdots x_d)$$

for all $(x_1, \cdots, x_{k-1}, x_{k+1}, \cdots, x_L) \in [0,L]^{d-1}$ and all $k = 1, \cdots, d$.

# M    MEMNO PSEUDOCODE

In Figure 10 we provide pseudocode for the MemNO framework (Section 5.3) in PyTorch (Paszke et al., 2019).

```python
from typing import List
from einops import rearrange
import torch
import torch.nn as nn

class MemNO(nn.Module):
    '''
    Notation:
        B: batch size
        T: temporal dimension
        S: spatial dimension (1D)
        H: hidden dimension
    '''
    def __init__(self, markovian_layers: List[nn.Module], memory_layer: nn.Module, memory_position: int):
        '''
        Args:
            markovian_layers: List of nn.Module that maps inputs (B, S, H) to outputs (B, S, H).
            memory_layer: nn.Module that maps inputs (B, T, H) to outputs (B, T, H).
                This must be a causal and sequential model.
            memory_position: Position of memory layer in the model.
        '''
        super().__init__()
        self.markovian_layers = nn.ModuleList(markovian_layers)
        self.memory_layer = memory_layer
        self.memory_position = memory_position

    def forward(self, x: torch.Tensor) -> torch.Tensor:
        '''
        Args:
            x: encoded representation of the solution of the PDE (B, T, S, H)
                (i.e. [u_0, ..., u_{T-1}])
        Output:
            Prediction of the solution of the PDE at the next timestep (B, T, S, H)
                (i.e. [\hat{u}_1, ..., \hat{u}_{T}])
        '''
        B, T, S, H = x.shape
        x = rearrange(x, 'b t s h -> (b t) s h')
        for markov_layer in self.markovian_layers[:self.memory_position+1]
            x = markov_layer(x)
        x = rearrange(x, '(b t) s h -> (b s) t h', b=B)
        x = self.memory_layer(x)
        x = rearrange(x, '(b s) t h -> (b t) s h', b=B)
        for markov_layer in self.markovian_layers[self.memory_position+1:]:
            x = markov_layer(x)
        return rearrange(x, '(b t) s h -> b t s h', b=B)
```

Figure 10: Pseudocode for the MemNO framework (see Section 5.3) in 1-D. Encoder and Decoder layers are omitted for clarity (see details in Appendix B).

