# OpenReview forum: "On the Benefits of Memory for Modeling Time-Dependent PDEs"
_ICLR.cc/2025/Conference — ICLR 2025 Oral_

### Official Review · Reviewer_akXT · 2024-10-16

**Soundness:** 3
**Presentation:** 3
**Contribution:** 3
**Rating:** 8
**Confidence:** 3

**Summary:**

This work applies a state space model to model the memory term of a PDE that (under assumption) decomposed by the Mori-Zwanzig theory. The method is mostly well-motivated and demonstrates improvement in solving PDEs in low resolution. Overall the paper addresses an important issue in the field of neural operators.

**Strengths:**

1. The paper is theoretically well motivated.
2. The proposed method for memory operators can be implemented easily.
3. The method demonstrates improvement in performance for solving PDEs.

**Weaknesses:**

1. The motivation for using S4 as the model of choice is not convincing.
2. Lack of numerical comparison against benchmarks in 2D cases.

**Questions:**

1. The main question I have is regarding the flexibility of the proposed method. The author showcased that the memory term modeled by S4 achieves improved performance, but in Appendix G, the memory term modeled by LSTM can be almost just as good. I hope the authors can also highlight this in the main text as I think this shows that adding the memory term is important, no matter the architecture.
2. Related to the above, can similar memory terms be added to the other benchmarks like U-Net neural operator? If so this can also be highlighted as an ablation study such that the importance of the memory term can be further recognized.
3. When the high-frequency mode matters, would it simply be more effective to increase the complexity of FFNO rather than applying the memory term? Can the author briefly discuss how this trade-off affects the choice of using a memory term (or not) in this case?
4. How well can the model perform for zero-shot super-resolution once the model is trained?

---

> ### Author Response · Authors · 2024-11-21
> **Response to Reviewer akXT**
>
> We find it encouraging that the reviewer found our work well motivated and the method to include memory easy to implement. Furthemore, we thank the reivewer for their suggestion on additional experiments and ablations on adding different Markovian operators and has led to two additional results: 1) we show that MemNO framework also improves the performance with the U-Net model as the Markovian model (Appendix G.3), and 2) S4FFNO still outperforms an FFNO model that is 7x bigger (Appendix H.2)!
> Please find our responses to your questions below:
>
>
> > The main question I have is regarding the flexibility of the proposed method. The author showcased that the memory term modeled by S4 achieves improved performance, but in Appendix G, the memory term modeled by LSTM can be almost just as good. I hope the authors can also highlight this in the main text as I think this shows that adding the memory term is important, no matter the architecture.
>
> Yes! The key motivation of our work is to show *when* memory is helpful for modeling PDEs, and not a comment on *which* architecture should be used to model memory. We believe that having many different architectures with memory outperforming Markovian ones reinforces our claim.  We explicitly included this point in Section 6.1 of the updated PDF (lines 431-434) and the Conclusion (lines 536-538).
>
>
>
> > Related to the above, can similar memory terms be added to the other benchmarks like U-Net neural operator? If so this can also be highlighted as an ablation study such that the importance of the memory term can be further recognized.
>
> Following the suggestion of the reviewer, we have applied the MemNO framework to the U-Net architecture, and have found similar findings when perfoming the KS and Burgers experiments of Section 6.1 and Appendix C. We have included this experiment in Appendix G.3. Here we also present and analyze the results. The following table contains the nRMSE values in the same setup as Table 1:
>
> | Dataset Name   | Resolution | U-Net   | S4U-Net  |
> |----------------|------------|---------|----------|
> | KS \($\nu$=0.075\) | 32       | 0.542   | **0.364**    |
> | KS \($\nu$=0.075\) | 64       | 0.147   | **0.114**    |
> | KS \($\nu$=0.075\) | 128      | **0.033**   | 0.058    |
> | KS \($\nu$=0.1\)   | 32       | 0.511   | **0.277**    |
> | KS \($\nu$=0.1\)   | 64       | 0.062   | **0.052**    |
> | KS \($\nu$=0.1\)   | 128      | **0.027**   | 0.030    |
> | KS \($\nu$=0.125\) | 32       | 0.249   | **0.104**    |
> | KS \($\nu$=0.125\) | 64       | **0.022**   | 0.026    |
> | KS \($\nu$=0.125\) | 128      | **0.014**   | 0.022    |
> | Burgers'        | 32         | 0.188   | **0.096**   |
> | Burgers'        | 64         | 0.171   | **0.070**    |
> | Burgers'        | 128        | 0.112   | **0.057**    |
>
> **It can be seen that enhacing the U-Net operator with memory brings significant benefits when the resolution is low, with S4U-Net outperforming U-Net in Burgers' and KS with resolution 32, achieving sometimes more than 2x lower error.** However, in higher resolutions (128 in KS), the memory is not needed and S4U-Net suffers a degradation in performance. We hypothesize that with a more extensive tuning of the training hyperparamters this degradation can be minimized.
>
> This experiment reinforces our main message, which is that **including memory can be very helpful, regardless of the architecture.** Moreover, it shows the **flexibility of our proposed MemNO framework**, as it can be applied to U-Net without modifications. We thank the reviewer for pointing out this important ablation and have added the experiments to our paper (Section G.3 of the Appendix).

---

> ### Author Response · Authors · 2024-11-21
> **Response to Reviewer akXT (Continuation)**
>
> > When the high-frequency mode matters, would it simply be more effective to increase the complexity of FFNO rather than applying the memory term? Can the author briefly discuss how this trade-off affects the choice of using a memory term (or not) in this case?
>
> We do not think that increasing the complexity of FFNO can bridge the gap in performance with S4FFNO, because fundamentally FFNO is Markovian and does not use memory to compensate for the partial observation of high frequency modes. To illustrate this point, we increased the complexity of FFNO (hidden dimension and number of layers) in the KS experiment of Section 6.1 with viscosity $\nu=0.1$. The results are the following:
>
> | Architecture   | Hidden Dimension | # Layers | nRMSE (f=32) | nRMSE (f=48) | #Params (f=32) | #Params (f=48) |
> |--------------|---------|---------|------------|------------|-------------|-------------|
> | S4FFNO | 128     | 4       | **0.108**      | **0.045**      | **2.8**         | **3.9**         |
> | FFNO         | 128     | 4       | 0.440      | 0.238      | 2.8         | 3.8         |
> | FFNO         | 128     | 8       | 0.361      | 0.181      | 5.5         | 7.6         |
> | FFNO         | 256     | 4       | 0.435      | 0.252      | 11.1        | 15.3        |
> | FFNO         | 256     | 8       | 0.346      | 0.194      | 22.2        | 30.6        |
>
> **It can be seen that S4FFNO achieves a much better performance than FFNO (more than 3x smaller nRMSE), despite having up to 7x less parameters. Thus, the memory term is essential to achieve good performance, and the accuracy-efficiency of S4FFNO is much more favorable.**
>
> We also found this suggested experiment very valuable and have included in Appendix H.2 of the paper.
>
> > How well can the model perform for zero-shot super-resolution once the model is trained?
>
> When the training resolution is low, as in our experiments, it is very hard to perform zero-shot super resolution. We have tried zero-shot super resolution by keeping only the Fourier modes seen during training (as in the original FNO paper [1]), however both FFNO and S4FFNO have very poor performance in this case. For example, when the training resolution is 64, S4FFNO and FFNO obtain a nRMSE of 0.011 and 0.033, respectively. However, when evaluated at a slightly higher resolution of 80, their nRMSE go to over 0.3, which is an order of magnitude higher, which indicates that **zero-shot resolution is very difficult when there are high frequency components, regardless of memory.**
> We believe that the achieved super-resoltuion in the original FNO paper is possible in the regime where the solutions do not contain high frequency components. For example, they perform super resolution in the Burgers' experiment with viscosity $\nu=0.1$ by training at resolution 256 and evaluating at higher resolutions. We note that most of the frequency spectrum of this Burgers' PDE is contained Fourier modes well below 128, as we show in our Figure 6. Thus, the first modes of the discrete fourier transform do not vary much when the observation resolution is changed from 256 to higher ones (intuitively, the fourier spectrum can be estimated accurately with resolution 256). This allows zero-shot super resolution, since FNO works with only the top K modes.
> Nevertheless, in our experiments the solutions of the PDEs contain a large fraction of high frequency components. Therefore, the discrete fourier transform changes a lot when the observation resolution is changed.
>
> [1] Li, Zongyi, Nikola Kovachki, Kamyar Azizzadenesheli, Burigede Liu, Kaushik Bhattacharya, Andrew Stuart, and Anima Anandkumar. 2020. “Fourier Neural Operator for Parametric Partial Differential Equations.”

---

> ### Author Response · Authors · 2024-11-21
> **Response to Reviewer akXT (Continuation)**
>
> > (Weakness) The motivation for using S4 as the model of choice is not convincing.
>
> As observed by the reviewer, there may be other memory architectures that outperform Markovian ones. In this sense, we agree that S4 is not the *only* model that can incorporate memory, which we have emphasized in the updated version in the paper (lines 431-434). Nevertheless, S4FFNO shows strong performance against Markovian baselines and **serves to prove our point that memory helps when there is partial observability.** We also note that S4 models are have favourable (linear) scaling with the length of the input (as opposed to quadratic scaling with Transformers), which could be beneficial for long range modeling high dimensional PDEs, for example 3D turbulent flows. Empirically, we also found that LSTMs were unstable to train in our 2D Navier Stokes experiments, while S4 layers were stable to train and are consistently performant.
>
> > (Weakness) Lack of numerical comparison against benchmarks in 2D cases.
>
>
> We demonstrate the effectiveness of our approach on both 1D and 2D partial differential equations (PDEs), including the 2D Navier-Stokes equations. We perform experiments in the 2D case where we show the effects of adding observation noise to the input PDEs and how S4FFNO consistently outperforms FFNO. For furhter ablation studies we focus on the 1D PDE cases due to the significantly longer training times for 2D problems (approximately 5 times slower). We note that our ablations suggest that *even* in the 1D cases there are clear benefits of explicitly modeling the memory!

---

> > ### Comment · Reviewer_akXT · 2024-11-21
> >
> > Thank you for the response. I am glad that the authors found my suggestions helpful and I believe the additional experiments/ablation studies have further strengthened the work.

---

### Official Review · Reviewer_oCpa · 2024-11-03

**Soundness:** 2
**Presentation:** 3
**Contribution:** 3
**Rating:** 6
**Confidence:** 4

**Summary:**

This paper analyzes the advantages of incorporating memory into AI models for solving time-dependent partial differential equations (PDEs). Traditional AI-driven approaches often rely on a Markovian assumption, where future states are predicted solely from the current state. However, the authors argue that including memory (i.e., information from past states) improves model accuracy, particularly when solutions are low-resolution or contain noise.

The authors further present the Memory Neural Operator (MemNO), a novel architecture that combines state space models (namely S4) and Fourier Neural Operators (FNOs) to capture both spatial and temporal dynamics effectively. Theoretical foundations based on the Mori-Zwanzig theory suggest that memory-inclusive models can outperform Markovian models in specific scenarios, particularly with high-frequency Fourier components. Empirical results show MemNO achieves up to 6x lower test error compared to memoryless models.

**Strengths:**

* This paper seeks to formalize and theoretically analyze the role of memory in learning time-dependent PDEs, distinguishing itself from prior studies that have approached memory in this context primarily from an empirical standpoint.
* The experiments and ablations are thorough and well-executed. Additionally, the empirical results are presented using relative errors, which is exemplary and should be the standard in this field. Unfortunately, this point deserves emphasis, as many other studies overlook this.
* the paper is well written and clearly structured
* the paper introduces recent state-of-the-art state-space models to the field of learning PDE solutions

**Weaknesses:**

* The paper claims to improve learning the solution of time-dependent PDEs theoretically grounded in the
    the Mori-Zwanzig formalism. However, equation 6 of the paper clearly states that the 'memory term' (i.e., second summand on the right hand side of the equation) involves the differential operator of the PDE, i.e., a local spatial operator. But the proposed method does not take into account spatial interactions and applies the memory cell (S4 layer) to each element of the spatial dimension independently. This is clearly not covered by the formalism and should be stated very clearly. That being said, it would be interesting to incorporate spatial interactions via Graph Neural Networks acting on the nearest neighbors for instance.
* The experiments are interesting and extensive. However, there are some issues that need to be addressed. It's not clear up to which extend the performance increase is due to the memory or simply due to the very expressive S4 model. One good way to analyze this empirically would be to plot the performance of the proposed method on the y-axis, while on the x-axis the size of the look-back window is varied, i.e., how many past instances are fed into the memory cell. Currently, all available past instances are fed to the memory cell. If the performance does not change significantly by varying that, this would indicate that memory is not as important and the performance difference can be explained by the different architectures. Figure 7 in the appendix already suggests that the architecture choice influences the performance.
* A general weakness: Why do we need an advanced AI model with 5M parameters only to solve a simple 1D Burgers equation that can be solved in milliseconds with traditional methods? I acknowledge that the aim of the paper is to improve AI models applied to PDEs. However, it would still be important to draw some connections to traditional methods, which in particular for the experiments considered in this paper are vastly superior. A good starting point to look into this is the referenced paper by McGreivy and Hakim (2024).

**Questions:**

See weaknesses

---

> ### Author Response · Authors · 2024-11-21
> **Response to Reviewer oCpa**
>
> We thank the reviewer for their detailed feedback, and are glad that the reviewer finds our theoretical results valuable and that our experiments are well executed. Furthermore, we thank the reivewer for their suggestion on conducting ablation experiments that compare the outputs of varying window lenghts to understand the effects of memory, which we have added in Section H.1 of the Appendix, as well as for pointing out to an important observation regarding the modeling of spatial information, which has led to Appendix I.
>
> Please find our response to the questions below:
>
>
> > The paper claims to improve learning the solution of time-dependent PDEs theoretically grounded in the the Mori-Zwanzig formalism. However, equation 6 of the paper clearly states that the 'memory term' (i.e., second summand on the right hand side of the equation) involves the differential operator of the PDE, i.e., a local spatial operator. But the proposed method does not take into account spatial interactions and applies the memory cell (S4 layer) to each element of the spatial dimension independently. This is clearly not covered by the formalism and should be stated very clearly. That being said, it would be interesting to incorporate spatial interactions via Graph Neural Networks acting on the nearest neighbors for instance.
>
> We agree that the memory term of the Mori-Zwanzig formalism includes spatial derivatives. We do not explicitly model such spatial information in the memory layer of the MemNO framework, but the hidden dimension can in principle encode this local information implicitly. We agree with the reviewer that this is an important observation which was not present in our original paper, so we have included it in the updated version (line 377 and Appendix I).
>
> Moreover, following the suggestion of the reviewer, we have tried several modifications to the S4FFNO architecture to provide an inductive bias for modeling spatial information more explicitly. Concretely, we tested the following architecture modifications:
>  - **S4FFNO + Input Gradients**: The numerical gradients of the input $u_t$ are fed to the model encoder. Specifically, $u_t$ is represented as a vector of spatial shape $S$, and the spatial gradients are approximated using second-order accurate central differences method (using the method `torch.gradient` of the `torch` library). This would directly facilitate storing information about the first order spatial gradients in the hidden dimension.
>  - **S4FFNO + Convolutional Encoder**: Instead of having a linear layer as encoder, we substitute it with a 1D Convolution of kernel size 3 across the spatial dimension.
>  - **S4FFNO + Convolution before memory layer**: Before feeding the sequence of hidden presentations $[v_0, v_1, ..., v_t]$ to the memory layer S4, we apply a 1D Convolution  across the spatial dimension with kernel size of 3. This is similar to modeling local interactions through Graph Neural Network, as the reviewer suggested, yet in a simple manner where each hidden dimension node only has two neighbours (its adjacent spatial points).
>
>  We use the setup of the KS experiment with viscosity $\nu=0.1$ and resolutions $f=16, 32, 48, 64$ and $80$, and present the nRMSE values in the following table:
>
> | Architecture |f=16      | f=32      | f=48      | f=64      | f=80      |
> |----------------------------------------------|---------|---------|---------|---------|---------|
> | FFFNO | 1.106 | 0.4461 | 0.2325 | 0.328 | **0.0040**
> | S4FFNO (Base)                                      | **0.3318**  | 0.1081  |**0.0455**  | **0.0111**  | 0.0048  |
> | S4FFNO + Input Gradients                     | 0.4433  | 0.1120  | 0.0482  | 0.0120  | 0.0054  |
> | S4FFNO + Convolutional Encoder               | 0.4340  | **0.1053** | 0.0571  | 0.0138  | 0.0065  |
> | S4FFNO + Convolution before memory layer       | 0.4283  | 0.1224  | 0.0491  | 0.0142  | 0.0057  |
>
> It can be seen that none of the methods provides better performance than S4FFNO, which we believe is because training becomes more difficult with these architectures. It would a great direction for future work to consider other ways to explicitly model spatial information and further improve the performance of memory architectures. We have included this experiment in Appendix I.

---

> ### Author Response · Authors · 2024-11-21
> **Response to Reviewer oCpa (Continuation)**
>
> > The experiments are interesting and extensive. However, there are some issues that need to be addressed. It's not clear up to which extend the performance increase is due to the memory or simply due to the very expressive S4 model. One good way to analyze this empirically would be to plot the performance of the proposed method on the y-axis, while on the x-axis the size of the look-back window is varied, i.e., how many past instances are fed into the memory cell. Currently, all available past instances are fed to the memory cell. If the performance does not change significantly by varying that, this would indicate that memory is not as important and the performance difference can be explained by the different architectures. Figure 7 in the appendix already suggests that the architecture choice influences the performance.
>
> Regarding the concern about whether the superior performance of S4FFNO comes from modeling memory or simply from the expressivity of the S4 layer, we performed the suggested experiment. In the KS experiment with viscosity $\nu=0.1$ and resolution 32 (same setting as in Section 6.1), we used different temporal window lengths for the memory model S4: T=0, T=5, T=10, T=15, T=20 and T=25 (the model always predicts 25 timesteps in total, but uses the memory of the past T previous timesteps to predict the next one). **It can be seen that the performance improves from having no memory (similar to FFNO) to having 25 timesteps of memory (4x lower error). Thus, the improved performance is related to the amount of past memory information provided to the model.**
> |           | FFNO | S4FFNO (T=0)| S4FFNO (T=5) | S4FFNO (T=10) | S4FFNO (T=15) | S4FFNO (T=20) | S4FFNO (T=25) |
> |-----------|------|------|--------------|---------------|---------------|---------------|---------------|
> | nRMSE     |0.446 | 0.462| 0.370        | 0.353         | 0.263         | 0.182         | **0.110**     |
>
>
> Additionally, we would like to point out to another experiment we included in Appendix H.2. In that experiment, instead of decreasing the window length of S4FFNO, we increased the number of parameters of FFNO (up to 8 layers and 256 hidden dimension). That makes FFNO make ~7x more parameters than S4FFNO. Nevertheless, FFNO has 3x larger nRMSE (Table 4). **This indicates that the differences in performance cannot be explained from model expressivity alone, but rather about the inputs that are available to the model.**
>
> > A general weakness: Why do we need an advanced AI model with 5M parameters only to solve a simple 1D Burgers equation that can be solved in milliseconds with traditional methods? I acknowledge that the aim of the paper is to improve AI models applied to PDEs. However, it would still be important to draw some connections to traditional methods, which in particular for the experiments considered in this paper are vastly superior. A good starting point to look into this is the referenced paper by McGreivy and Hakim (2024).
> >
>
> **While our work does not directly offer a solution to the issues found by McGreivy and Hakim (2024), we believe it is a valuable contribution towards desigining neural operators that improve their accuracy-efficiency tradeoff.** S4FFNO was specifically designed to have an efficiency close to FFNO (see Table 2), yet **we found that it can achieve up to 6x lower test error than FFNO with only an ~14% increase in computation time.** Moreover, as we mentioned in another experiment we found that **S4FFNO has more than 3x lower test error compared to an FFNO model that is 7x bigger** (Table 4). Thus, in this sense we have provided a model with a better accuracy-efficiency tradeoff. More importantly, we provide an explanation of *when* using memory is necessary. This explanation can help the design of more cost-efficient architectures based on whether memory is required or not, especially given the community's push towards training large scale foundation models for complex high dimensional PDEs. We definitely agree with the sentiment that we need more challenging and meaningful benchmarks for machine-learned PDE solvers more broadly.

---

> > ### Comment · Reviewer_oCpa · 2024-11-22
> > **Thanks for the reply**
> >
> > I thank the reviewers for their response. My main point of criticism still remains, that the paper is motivated by heuristics rather than rigorous theory (i.e., ignoring spatial effects in the Mori-Zwanzig formalism). I thus keep my score of 6, which I consider an appropriate rating of this paper.

---

### Official Review · Reviewer_vkVf · 2024-11-04

**Soundness:** 3
**Presentation:** 3
**Contribution:** 3
**Rating:** 8
**Confidence:** 4

**Summary:**

The authors propose a variant of a factorized Fourier neural operator with memory and show how it can improve over the standard Markovian neural operator for modeling PDEs with small grid sizes or noise added to the observations. Additionally, some theory motivated by the Mori-Zwanzig theory of model reduction is given to motivate and explain the success of the addition of memory (Eq. 6). This success in certain regimes is observed in experiments.

**Strengths:**

1) The problem of whether to use memory to model time-dependent PDEs is well-defined and well-motivated.
2) The theoretical motivation (Eq. 6) and example (Section 4) are clear, concise, and show the possible situations where adding memory in Neural Operators could improve performance.
3) The two experiments varying resolution and additive noise nicely demonstrate regimes where memory can be helpful. Figure 6 also provides good context in this regard.

**Weaknesses:**

1) There should be a brief discussion about the algorithmic cost of adding memory to the FNO. The authors use the diagonal matrix parameterization of S4, so this should be a reasonable increase in cost. Empirical timing would also be helpful to make this point.
2) In the added noise experiments (Section 6.2) it is expected that memory improves performance because the memory unit can approximately filter the input data. The discussion surrounding this experiment is less fleshed out than that of the resolution experiments in Section 6.1 and does not have an obvious connection to the theory and example presented in sections 3 and 4. What practical situation is this intended to model?
3) lines 1119-1121: this is a strange training setup, but maybe necessary due to the GPU memory limitations.

**Questions:**

See Weakness 2.

---

> ### Author Response · Authors · 2024-11-21
> **Response to Reviewer vkVf**
>
> We are encouraged to see that the reviewer finds that our empirical and theoretical results are concise and clearly show the regimes where using memory is helpful for neural operators. Please find our responses to the questions below:
>
>
> > There should be a brief discussion about the algorithmic cost of adding memory to the FNO. The authors use the diagonal matrix parameterization of S4, so this should be a reasonable increase in cost. Empirical timing would also be helpful to make this point.
>
> Thank you for your suggestion, we agree that a discussion on training cost will help better understand the performance of different models! We provide a table with the empirical training times (forward + backward) of the different architectures for a batch size of 32, resolution 64 and 25 timesteps (the rest of the architecture details are the same as in Section 6.1), as well as the parameter counts. **We note that S4FFNO is only ~14% slower than FFNO, yet it achieves a loss that is 3x lower in the KS experiment with viscosity $\nu=0.075$ (see Table 1).** Moreover, we have included another experiment in Appendix H.2, where we show that **S4FFNO achieves 3x lower error than an FFNO model that has 7x more parameters (see Table 4).**
>
> | Architecture          | # Params (millions) | Training time (milliseconds) |
> |-----------------------|----------------------|-------------------------------|
> | Factformer (1D)       | 0.65                | 102                           |
> | GKT                   | 0.29                | 21                            |
> | U-Net                 | 2.68                | 23                            |
> | FFNO                  | 4.89                | 28                            |
> | Multi Input FFNO      | 4.89                | 28                            |
> | S4FFNO                | 4.94                | 32                            |
>
>
> We have included these values in Table 2 of appendix B.1. We used a NVIDIA RTX L40S for these measurements.
>
> As for the theoretical algorithmic complexity, S4FFNO has a smaller cost than FFNO. Let $S$ be the spatial resolution, $T$ the number of timesteps, $H$ the hidden dimension and $N$ the state dimension of S4. The core spectral convolution of FFNO (Eq. 16) has a Discrete Fourier Transform across the space dimension and a matrix multiplication in the frequency space, which have complexities $O(T H S \text{log}S)$ and $O(T S H^2)$ respectively. In contrast, the S4 layer has a Discrete Fourier Transform across the time dimension and it requires building the convolution kernel, which have complexities $O(S H T \text{log} T)$ and $O(S H (N+T) \text{log}(N+T))$ respectively (see [1] for details). In our cases, $S$ ranges from 32 to 128, and $T$ is either 20, 25 or 32, so  $O(T H T \text{log}T) < O(T H S \text{log}S)$. As for the other term, we use $H=128$ and $N=64$, thus we also have $O(S H (N+T) \text{log}(N+T)) < O(T S H^2)$. Thus, in theory the added S4 model to S4FFNO contributes to a factor smaller than one layer of FFNO to the total architecture. We have included this analysis in Appendix B.2.
>
>
> > In the added noise experiments (Section 6.2) it is expected that memory improves performance because the memory unit can approximately filter the input data. The discussion surrounding this experiment is less fleshed out than that of the resolution experiments in Section 6.1 and does not have an obvious connection to the theory and example presented in sections 3 and 4. What practical situation is this intended to model?
>
> This experiment is concerned with the practical situation where a physical measurement device has observation noise. We believe this is a situation of practical importance since **the measurement devices generally come with an observation error, especially when the observation grid is very large** (as in the case of 2D systems), where obtaining accurate measurements is very costly. In this case, we showed that using memory helps the neural operator. We checked that the observation resolution (64x64) is high enough to observe all relevant Fourier modes, and thus it is a different setting than Section 6.1. But still, memory helps the neural operator. We believe this reinforces one of our main claims, which is that **memory helps when the solution of the PDE is partially observed.**
>
>
> > lines 1119-1121: this is a strange training setup, but maybe necessary due to the GPU memory limitations.
>
> Indeed this setup was due to GPU memory limitations. When the training length was split into two chunks of length 16, the GPU memory requirement was 34GB for this experiment. Using the full length for training would require a memory beyond 48GB, which was beyond our GPU capacity (we used NVIDIA A6000/A6000-Ada GPUs). We have made this fact more explicit in our paper.
>
> [1] Albert Gu, Karan Goel, and Christopher Ré. Efficiently modeling long sequences with structured state spaces. In The International Conference on Learning Representations (ICLR), 2022

---

> > ### Comment · Reviewer_vkVf · 2024-11-21
> >
> > Thank you for your detailed and informative response. My concerns have all been addressed and the additional details added in response to the other reviewers' comments have strengthened the paper. I have raised my score from a 6 to an 8 as a result.

---

### Official Review · Reviewer_8n1K · 2024-11-04

**Soundness:** 3
**Presentation:** 3
**Contribution:** 3
**Rating:** 8
**Confidence:** 3

**Summary:**

The paper proposes a new form of neural operators (NOs) that are inspired by the Mori–Zwanzig (MZ) formalism. This operator is to model spatio-temporal dynamics that arise in time-dependent (nonlinear) partial differential equations (PDEs). The paper points out that the previous roll-out-of-NOs which is caused by memorylessness and, thus, proposes to implement the memory term similarly in the MZ formalism. The specific example architecture has been investigated: Fourier NO + S4, which are empirically shown to be more effective in capturing the dynamics. Moreover, the paper points that the previous benchmark PDEs are easy (including relatively only frequency) and, thus, proposes a new set of benchmark datasets that gives challenge to previous NOs.

**Strengths:**

The paper is well-motivated and proposes solution methods as well as a new set of datasets.

The paper is well-written, providing necessary details without going into too much detail and explaining why including the memory term can be beneficial in modeling complex dynamics.

The paper introduces a new challenging dataset (although not extensive), the dataset could be used as a next important benchmark for future works.

The empirical results seem to show that the proposed method is more effective in modeling the dynamics. Interestingly, the results are somewhat aligned with the effect of the MZ formalism in heterogeneous systems; that is, on a coarse mesh (on a coarse-grained or homogenized system in MZ), the Markovian surrogate model can be aided by the memory term to capture the influence the state defined on a finer mesh (the eliminated variables in MZ).

**Weaknesses:**

It is not easy to see exactly how the architecture is designed. Including diagrams showing the forward computational path will be very helpful.

Although taking a cue from the MZ formalism is good, some discussions on the related topic such as closure modeling for surrogate models seems to be missing.

FNOs are known to select low-frequency modes and are expected to struggle with high-frequency solutions. Are transformer-based neural operators expected to behave similarly? Could the authors provide why GKT performs poorly?

Also, could the authors provide experimental results with some advanced transformers-based neural operators? such as OFormer (Li, et al, TMLR, 2023) and their variants (for example, FactFormer, Li et al, NeurIPS 2023, 2023)

**Questions:**

Please see the questions in the weaknesses above.

---

> ### Author Response · Authors · 2024-11-21
> **Response to Reviewer 8n1K**
>
> We are pleased that the reviewer finds our work well-motivated and well-written, and that they find it interesting that the empirical results are aligned with the Mori Zwanzig theory, which we believe is a key insight of our work. We provide responses to their questions:
>
> > It is not easy to see exactly how the architecture is designed. Including diagrams showing the forward computational path will be very helpful.
>
> We appreciate the suggestion of adding a diagram. We have included it in Figure 9, along with the pseudocode for MemNO in Figure 10, which we hope can help facilitate the understanding of the architecture.
>
> > Although taking a cue from the MZ formalism is good, some discussions on the related topic such as closure modeling for surrogate models seems to be missing.
>
> We agree with the reviewer that there are other ways to model the unresolved dynamics beyond the Mozi-Zwanzig formalism. In our work, we explain *when* modeling the memory term of the Mori-Zwanzig equation is expected to bring benefits, we propose a neural architecture to build surrogate models and we back up our study with empirical and theoretical claims. We hope that our methodology stimulates more interest in the topic of closure modeling more broadly, the design of neural surrogates and the study of when the closures are expected to outperform Markovian approximations.
>
> Additionally, based on this suggestion we have included more citations of works that discuss the closure of surrogate models in the last paragraph of Section 3.2 (see lines 189-195).
>
>
> > FNOs are known to select low-frequency modes and are expected to struggle with high-frequency solutions. Are transformer-based neural operators expected to behave similarly?
>
> Fundamentally, we believe that all Markovian operators will struggle when the solution of the PDE has high frequency components beyond the observation resolution. In our experiments, Markovian operators take inputs on a low resolution grid, which cannot correctly capture high frequency modes. Thus, learning the solution operators exactly is not possible because the operator does not have information on the full spectrum. In the original GKT paper, it is also stated that GKT is not expected to perform well in the presence of high frequency components (see the Conclusion in [1]). In contrast, the Mori-Zwanzig formalism states that we can compensate for this lack of information by leveraging the memory of past states (Equation 6). **Thus, we generally expect Markovian operators to underperform their memory-augmented counterparts, as we observe in Table 1.**
>
> [5] Shuhao Cao. Choose a transformer: Fourier or Galerkin. Advances in Neural Information Processing Systems (NeurIPS 2021), 34, 2021.

---

> ### Author Response · Authors · 2024-11-21
> **Response to Reviewer 8n1K (Continuation)**
>
> > Could the authors provide why GKT performs poorly?
>
> In our experiments, it is the case that GKT has worse performance than FFNO, especially at resolutions 36-64 in Figure 1. In general, we found GKT harder to train, but after extensive finetuning we were able to improve its performance in this range of resolutions by making two modifications: 1) adding 0.05 dropout in the linear attention layer and 0.025 dropout in the FFN layer; and 2) reducing the training epochs from 200 to 50. However, we note that this change made the performance drop considerably in resolutions higher than 64. We present the results in the following table:
> | Resolution | GKT (Base)  | GKT (Dropout & 50 epochs) | $\Delta$(Base - Dropout & 50 epochs)  |
> |------------|-------|---------|--------|
> | 24         | 0.674 | 0.666   | 0.008  |
> | 28         | 0.677 | 0.636   | 0.041  |
> | 32         | 0.601 | 0.596   | 0.006  |
> | 36         | 0.595 | 0.563   | 0.032  |
> | 40         | 0.700 | 0.516   | 0.184  |
> | 44         | 0.760 | 0.499   | 0.261  |
> | 48         | 0.694 | 0.486   | 0.207  |
> | 52         | 0.630 | 0.427   | 0.202  |
> | 56         | 0.584 | 0.366   | 0.218  |
> | 60         | 0.331 | 0.258   | 0.073  |
> | 64         | 0.120 | 0.170   | -0.051 |
> | 68         | 0.049 | 0.134   | -0.085 |
> | 72         | 0.034 | 0.151   | -0.117 |
> | 76         | 0.024 | 0.125   | -0.102 |
> | 80         | 0.017 | 0.130   | -0.114 |
>
> In order to strenghten the baseline, we have included this hyperparameter sweep over the dropout in our experiment (Section 6.1), and have chosen the performance of the best model (with or without dropout). Although the GKT curves in Figure 1 are now smoother, the performance of FFNO is still better.
>
> However, we do not believe this is necessarily a limitation of Transformer-based neural operators. As the reviewer also suggested, we provide results for Factformer in the answer to the next question. We show that Factformer has superior performance than FFNO in lower resolutions, although worse performance in higher resolutions.

---

> ### Author Response · Authors · 2024-11-21
> **Response to Reviewer 8n1K (Continuation)**
>
> > Also, could the authors provide experimental results with some advanced transformers-based neural operators? such as OFormer (Li, et al, TMLR, 2023) and their variants (for example, FactFormer, Li et al, NeurIPS 2023, 2023)
>
> **We haved provided results for Factformer and included it in the paper as an additional baseline** for the experiments in Section 6.1 (KS) and Appendix B (Burgers'). The results are shown in Table 1 and Figure 1. Here we also present the results for Factformer, GKT and S4FFNO to facilitate the comparison:
> | Dataset      | Resolution | Factformer (1D) | GKT     | S4FFNO (Ours) |
> |--------------------|------------|-----------------|---------|---------------|
> | KS ($\nu=0.075$) | 32         | 0.436           | 0.588   | **0.139**     |
> | KS ($\nu=0.075$) | 64         | 0.195           | 0.401   | **0.036**     |
> | KS ($\nu=0.075$) | 128        | 0.058           | 0.028   | **0.008**     |
> | KS ($\nu=0.1$)   | 32         | 0.391           | 0.601   | **0.108**     |
> | KS ($\nu=0.1$)   | 64         | 0.086           | 0.120   | **0.011**     |
> | KS ($\nu=0.1$)   | 128        | 0.030           | 0.013   | **0.005**     |
> | KS ($\nu=0.125$) | 32         | 0.149           | 0.314   | **0.031**     |
> | KS ($\nu=0.125$) | 64         | 0.022           | 0.016   | **0.004**     |
> | KS ($\nu=0.125$) | 128        | 0.017           | 0.007   | **0.003**     |
> | Burgers'           | 32         | 0.165           | 0.356   | **0.053**     |
> | Burgers'           | 64         | 0.144           | 0.349   | **0.037**     |
> | Burgers'           | 128        | 0.106           | 0.307   | **0.030**     |
>
> Factformer outperforms GKT in low resolutions of KS and in the Burgers' equation, yet has worse performance in high resolutions. As seen in Figure 1, Factformer is a strong Markovian baseline in low resolutions, yet it is still outperformed by our memory model S4FFNO. **We believe that showing the superiority of S4FFNO against an even stronger Markovian baseline reinforces our claim that memory helps in low resultion.**
>
> As for the difference between GKT and Factformer, a possible explanation for the superior performance of Factformer in low resolutions might be related to how GKT and Factformer introduce positional information for the transformer layer.  GKT encodes position by concatenating the grid into the queries, keys and values; while Factformer introduces position through cross attention and additionally has an additive positional encoding. This latter method dedicates more parameters to the positional information, which might be helpful in low resolution to give a different treatment to each spatial position and compute a better approximation of the Fourier spectrum. In contrast, in higher resolutions this might be too fine-grained information and Factformer fails to generalize.
>
> Nevertheless, there are several other confounders in the architecture. For example, GKT uses a FNO-based regressor after the transformer encoding layers, while Factformer uses an MLP regressor. Additionally, Factformer uses an MLP to build the queries and keys, while GKT only uses a linear projection. **We hope that our proposed benchmark can illustrate new tradeoffs between architectures and can help design better ones that achieve high performance in both low and high resolutions.**
>
> Lastly, we mention that we also implemented Oformer (taking it from the repository https://github.com/BaratiLab/OFormer). However, in our KS experiments with low viscosities and low resolutions, the test loss diverged because rolling out Oformer for 25 timesteps lead to very large values. Just like GKT and Factformer, Oformer was tested in a different experimental setup and has several architectural differences (for example, in [3] Oformer is not rolled out for several timesteps from a single initial condition in 1D experiments, while in [2] Factformer was only applied to 2D problems). We hypothesize that making architectural changes would solve the training instabilities. But given that the architecture of Factformer is similar to Oformer, we believe that having Factformer as a baseline is already representative of the performance of more advanced transformer architectures.
>
>
> [2] Li, Z., Shu, D., & Barati Farimani, A. (2023). Scalable Transformer for PDE Surrogate Modeling
>
> [3] Li, Zijie, Meidani, Kazem, and Barati Farimani, Amir. "Transformer for Partial Differential Equations’ Operator Learning." Transactions on Machine Learning Research, 2023.

---

> > ### Comment · Reviewer_8n1K · 2024-11-22
> >
> > Thank you very much for detailed response; the comments on Markovian operators cleared my comments on Transformer-based neural operators. Also, I appreciate the new experimental results. I adjusted my score accordingly.

---

### Author Response · Authors · 2024-11-21
**General response to reviews and updated PDF**

We thank reviewers for their feedback! We are encouraged to see that the reviewers find our study on the benefits of memory on neural operators important and well motivated [8n1K, vkVf, oCpa, akXT], and find our empirical analysis and ablations to be thorough [oCpa,vkVf]. Similarly, we are glad that the reviewers appreciated our theoretical treatment of memory based  on the Mori-Zwanzig equation, which distinguishes us from prior works and is strongly backed up by our experiments [8n1K, vkVf, oCpa]. Lastly, we appreciate that the reviewers found valuable our new challenging benchmark of PDEs in low resolutions [8n1K] and our proposed MemNO as an effective and easy to implement architecture to model memory [akXT].

Based on the suggestions of the reviewers, we have modified the original version of the paper with more baselines, experiments and ablations. We believe these new results reinforce our main claims of the paper:
 - Our S4FFNO has much superior performance than another newly introduced baseline (Factformer) and a more extensively finetuned GKT baseline, showing that **modeling memory brings significant performance gains compared to an even wider range of Markovian baselines.** *(Table 1 and Figure 1)*
 - The MemNO framework with U-Net as the Markovian model and S4 as the memory model outperforms the purely Markovian U-Net in low resolutions. **This illustrates the flexibility of our framework and shows that the benefits of memory do not apply to only one specific architecture.** *(Table 4)*
 - S4FFNO outperforms the Markovian FFNO by a large margin even when FFNO has 7x more parameters, proving that **the superior performance of S4FFNO comes from the ability to model memory**. *(Table 5)*
 - S4FFNO improves performance as it has access to a larger memory window, again showing that **the performance gap between S4FFNO and FFNO is due to the memory of the full history of previous timesteps.** *(Figure 8)*

The changes in the updated version are in green, we briefly summarize them here:
1. **Factformer as an additional baseline:** We included Factformer [1] as an additional Markovian baseline in our KS and Burgers' experiments (Sections 6.1 and C, respectively). The results are shown in Table 1 and Figure 1. **Our proposed S4FFNO architecture outperforms Factformer in low resolutions**. Details on the architecture are given in Appendix B.
4. **S4U-Net as another effective memory model**: We showed the flexibility of our proposed MemNO framework by combining S4 as the memory model and U-Net as the neural operator. As in the case of FFNO, **S4U-Net outperforms U-Net in low resolutions**, see Appendix G.3. Thus, this experiments shows that the benefits of memory are not unique to a specific archiecture, and the MemNO framework is effective when applied to other neural operators apart from FFNO. We further emphasized this point in our paper by explaining that several memory architectures are able to outperform Markovian ones (lines 431-433), which we believe reinforces our main contributions regarding the effect of memory.
5. **Comparison between S4FFNO and FFNO with up to 7x more parameters**: We conducted an experiment (Appendix H.2) **where we increased the model size (hidden dimension and number of layers) of FFNO up to 7x times, and still the original S4FFNO outperformed it and achieved 3x less error**.
6. **Memory window length of S4FFNO:** We show that S4FFNO increases performance as it has access to the memory of more previous timesteps, going from the FFNO performance (no previous timesteps) to 4x lower error. This eliminates confounding factors and indicates that **memory is the reason for the improved performance of S4FFNO**. The ablation is showed in Appendix H.1.
7. **Computational cost and algorithmic complexity of neural operators:**  We provided empirical timings for all architectures, as well as an analysis on the algorithmic complexity of FFNO and S4FFNO. The results are shown in Appendix B.1 and B.2.
8. **Improved GKT baseline**: The Markovian GKT baseline was strenghtened by more extensive hyperparameter tuning (Figure 1). The details on the training setup are included in Appendix B.1. We note that these new hyperaparameters do not improve the performance on the results previously shown in Table 1, and still S4FFNO and other Markovian baselines outperform GKT.
9. **References on closure modeling in Mori-Zwanzig discussion**: We included references related to the topic of closure modeling of surrogate models in Section 3.2 (lines 189-194).
10. **Ablation on the modeling of local spatial information**: On Appendix I, we have included an ablation on architecture modifications that model spatial information more explicitly to approximate the memory term.
11. **Diagram of MemNO**: We have included a diagram of the MemNO framework in Figure 9.


[1] Li, Z., Shu, D., & Barati Farimani, A. (2023). Scalable Transformer for PDE Surrogate Modeling

---

### Meta-Review · Area_Chair_7vp4 · 2024-12-20

**Metareview:**

This paper introduces a novel neural operator with memory for modeling time-dependent PDEs. By combining state-space models and Fourier neural operators, the authors establish a sound framework that extends beyond standard Markovian neural operators. The approach is not only novel, but also demonstrates its significance through strong experimental results. In addition, Mori-Zwanzig theory provides a theoretical motivation for the proposed model. Overall, the key strengths of this paper lie in its innovative architecture, theoretical underpinnings, and strong empirical findings.

All reviewers agree that this is a very strong submission. In particular, I believe that this paper has the potential to inspire interesting future research. Thus, I strongly recommend accepting this paper.

**Additional Comments On Reviewer Discussion:**

The authors have used the rebuttal phase to address the reviewers’ concerns, adding new baselines, experiments, and ablation studies.

---

### Decision · Program_Chairs · 2025-01-22

Accept (Oral)